# Molecular basis for DNA recognition by the maternal pioneer transcription factor FoxH1

Radoslaw Pluta [1,8], Eric Aragón[1,8], Nicholas A. Prescott [2,3], Lidia Ruiz[1], Rebeca A. Mees [1], Blazej Baginski [1], Julia R. Flood[3], Pau Martin-Malpartida [1], Joan Massagué [4], Yael David [3,5,6] & Maria J. Macias [1,7] ✉

Forkhead box H1 (FoxH1) is an essential maternal pioneer factor during embryonic development that binds to specific GG/GT-containing DNA target sequences. Here we have determined high-resolution structures of three FoxH1 proteins (from human, frog and fish species) and four DNAs to clarify the way in which FoxH1 binds to these sites. We found that the protein-DNA interactions extend to both the minor and major DNA grooves and are thus almost twice as extensive as those of other FOX family members. Moreover, we identified two specific amino acid changes in FoxH1 that allowed the recognition of GG/GT motifs. Consistent with the pioneer factor activity of FoxH1, we found that its affinity for nucleosomal DNA is even higher than for linear DNA fragments. The structures reported herein illustrate how FoxH1 binding to distinct DNA sites provides specificity and avoids cross-regulation by other FOX proteins that also operate during the maternal-zygotic transition and select canonical forkhead sites.

FoxH1 (also known as Fast1) belongs to the forkhead family of transcription factors[1,2]. It was initially identified as a mediator of activin signaling in Xenopus embryos[3]. *FoxH1* corresponds to the *schmalspur* (*sur*) locus in zebrafish and mutations are associated with defects in dorsal midline development in zebrafish and mice[4,5]. Basal binding of FoxH1 primes mesendoderm differentiation promoters to activate transcription[6]. FoxH1 has also been identified as the earliest maternal pioneer factor[7] participating in the coordination of zygotic genome activation[8,9]. At the blastula stage, which is characterized by relatively nucleosome-dense chromatin, FoxH1 functions at the top of the regulatory hierarchy and marks super-enhancers for the activation of developmental genes[6,8,10–12]. At the start of gastrulation, FoxH1 is key to transducing activin/nodal signals, which operate through the transforming growth factor beta (TGFβ) signaling pathway[13,14] and participate in maternal regulation of Wnt/β-catenin target genes for differentiation[6,15–17]. The essential roles played by FoxH1 are underscored by the identification of *FoxH1* mutants that lack anteroposterior axis specification, primitive streak patterning, and heart development[18,19].

The members of the FOX transcription factor family share the presence of the forkhead domain (FH), which interacts with DNA (InterPro entry IPR001766). Most FOX proteins recognize the canonical forkhead DNA motif TRT**TT**RY (R = A/G, Y = C/T)[2–4,20,21]. In contrast, FoxH1 recognizes the TGT**G(G/T)**ATT sequences in several cis-regulatory elements, such as in *Gsc*, *Eomes*, *Nodal*, *Mixl*, and *FoxA2*, to confer activin regulation in vivo[6,8,16,22–24].

Although many structures of other winged helix factors bound to forkhead DNA motifs have been reported[2,21], there is a dearth of structural information regarding FoxH1 recognition of its specific DNA sites. Understanding these distinct specificities is essential to define the roles of FoxH1 in embryo development and in tumors associated with elevated expression FoxH1 in humans[25–27]. To fill this knowledge

[1]Institute for Research in Biomedicine (IRB Barcelona), The Barcelona Institute of Science and Technology (BIST), Barcelona 08028, Spain. [2]Tri-Institutional PhD Program in Chemical Biology, New York, NY, USA. [3]Chemical Biology Program, Memorial Sloan Kettering Cancer Center, New York, NY 10065, USA. [4]Cancer Biology and Genetics Program, Memorial Sloan Kettering Cancer Center, New York, NY 10065, USA. [5]Department of Pharmacology, Weill Cornell Medicine, New York, NY 10065, USA. [6]Department of Physiology, Biophysics and Systems Biology, Weill Cornell Medicine, New York, NY 10065, USA. [7]Institució Catalana de Recerca i Estudis Avançats (ICREA), Passeig Lluís Companys 23, Barcelona 08010, Spain. [8]These authors contributed equally: Radoslaw Pluta, Eric Aragón. ✉e-mail: maria.macias@irbbarcelona.org

gap, we selected FH domains from the FoxH1 proteins of three species (*Homo sapiens*, *Xenopus laevis*, and *Danio rerio*) and determined the crystal structures of these domains bound to the TGT**GG**ATT dsDNA sequence from the native promoter of the Goosecoid protein (Gsc). We also characterized the FH domain complexes with TGT<u>GT</u>ATT, as well as with the TGT**TT**ATT variant and the canonical forkhead motif (TRT**TT**RY) to determine the key aspects that define the specificity of FoxH1 with respect to other FOX proteins. These complexes revealed the presence of unique flanking regions surrounding the FH domain core, and how two specific amino acid differences in FoxH1 enhance its exceptional ability to bind to GG/GT (GK) sites. In particular, we observed a compact and well-structured Wing2 region—a form never previously observed in other FOX proteins.

While nucleosomes are barriers for the binding of other transcription factors, pioneer factors bind to nucleosome DNA to facilitate subsequent protein interactions for transcription activation. Consistent with the role of FoxH1 as a pioneer factor, binding assays with reconstituted mononucleosomes derived from the Widom 601 sequence and also from the *Gsc* promoter showed a slight preference for the FoxH1 FH domain for binding to nucleosome core particle over linear DNA. Collectively, these results explain that, although FoxH1 also interacts with canonical forkhead motifs in vitro, binding to TGT**GK**ATT target sites in vivo appears to be an essential property to avoid cross-regulation by other FOX proteins that also operate during embryonic development and select canonical forkhead sites only.

## Results

### The FoxH1 FH domain binds DNA with high affinity

In vertebrates, FoxH1 proteins have a FH DNA-binding domain followed by the engrailed homology 1 motif and the SMAD-interacting domain, the latter two involved in protein recognition (Fig. 1a)[28–30]. In the FoxH1 sequence, there is a short positively charged motif preceding the FH domain, and a C-terminal region that is highly conserved in vertebrates. We used JPredv4[31], a protein secondary structure prediction server, to analyze this C-terminal region and detect helical propensities with high confidence for an area comprising nearly 40 residues. Given its hydrophobic nature and conservation degree, we suspected that this C-terminal region might fold and interact with the FH core, giving rise to a larger FH domain than that present in other FOX proteins (Fig. 1b, Supplementary Fig. 1a).

To study the DNA-binding capacity of FoxH1, we selected a fragment of the *Gsc* promoter containing the TGTGGATT site (GG site), defined as a functional recognition site in the literature (Fig. 1c)[6,24], and the TGTGTATT variant, which is highly represented in ChIPseq peaks corresponding to loci bound by FoxH1 (Supplementary Fig. 1b). In addition, we analyzed the FoxH1 interaction with the TGT**TT**ATT site and with the canonical forkhead motif (TGTTTAC) for comparison with other FOX proteins, although these TT motifs scored lower than GG/GT (GK) sites in experimental assays (Fig. 1c, Supplementary Fig. 1b).

We expressed protein constructs of ~140–180 residues from three vertebrates and also a canonical FH domain that spans ~100 residues (Supplementary Fig. 1c). Using differential scanning fluorimetry, we observed that the most stable constructs were those that included extended boundaries on both sides of the FH domain. These constructs also showed greater changes in stability upon DNA binding (Fig. 1d). For the extended human and zebrafish FoxH1 constructs, a clear increase in the stability of the protein was detected for the TGTGKATT motifs (~17 °C), while this increase was less prominent for the TGTTTATT and TGTTTAC ones (14 °C and 11 °C). The construct spanning the canonical domain showed no increase in stability in the presence of DNA (Supplementary Table 1). In contrast, FoxA2, also a pioneer factor that binds to canonical forkhead motifs[32], displayed the opposite behavior and selected only for DNAs with TT sites, even

showing a decrease in its melting temperature in the presence of the GG site. A competition assay using full-length (FL) FoxH1 protein expressed in HEK293T cells and the recombinantly expressed extended domain indicated that the extended domain recapitulates the DNA binding capacities of FL-FoxH1 (Supplementary Fig. 1d). Similar results were previously observed when comparing recombinant GST-FL-FoxH1 protein and a slightly shorter FH domain[24].

The binding effects measured as temperature changes were corroborated by native gel electrophoresis and 16 bp DNA cy5-labeled fragments. These assays revealed FoxH1-DNA interactions in the nanomolar range. Also, EMSA assays indicated that FoxA2 binds weakly to the TGTGGATT site whereas it interacts with high affinity with its forkhead site. Using cell lysates and overexpressed FH domains, we performed competition assays between the FL-FoxH1 protein and the FH domain of FoxA2 using the forkhead site as the ligand. We observed that only high FL-FoxH1 concentrations could displace FoxA2 from the DNA (Fig. 1e, Supplementary Fig. 1d). These findings corroborate that FoxH1 and FoxA2 interact with TT sites, although the former has the distinct ability to recognize GK sites, a rare feature among FOX proteins. Our findings also indicate that the FoxH1 FH domain is larger than other FH domains of FOX proteins.

### Motif analysis: FoxH1 vs. FoxA proteins

To analyze whether the different DNA binding preferences we detected for FoxH1 and FoxA2 in vitro are identifiable in a native context, we used ChIP-seq data available in the Gene Expression Omnibus GEO databases for FoxH1 and FoxA proteins. In *Xenopus tropicalis*, during the maternal to zygotic transition, the level of FoxH1 increases (stages 8 and 9) and then decreases (stage 10.5) simultaneously with the increase in zygotic FoxA protein expression[22]. Considering these variations in protein concentration, for this analysis we used the available ChIP-seq entries for FoxH1 (stages 8, 9, and 10.5) and stage 10.5 for FoxA proteins[8,22].

When comparing motifs present in peaks of stages 8, 9, and 10.5 (FoxH1) and 10.5 (FoxA), using the Simple Enrichment Analysis (SEA) algorithm[33], we found motif enrichment for GG and GT sites (GK) at FoxH1 peaks at all stages, being particularly high at stage 10.5 (x7.31 enrichment), indicating that FoxH1 remains significantly bound to the GK sites even when the protein levels are low. In the case of FoxA proteins, the analysis shows a slight enrichment for the Forkhead motif, which is not enriched in FoxH1 ChIPseq peaks (Supplementary Fig. 1e). In agreement with the low affinity we detected in EMSA assays for FoxA2 and the GK sites, there is no enrichment in GK sites in FoxA ChIPseq peaks. Intriguingly, when we focused the analysis on the common regions, (either FoxH1∩FoxA or FoxH1∩FoxA, Supplementary Fig. 1e), we observed a similar enrichment for GK sites as in FoxH1 peaks whereas there is a small decrease in Forkhead sites and an increase in TT sites relative to the enrichment measured for the full set of FoxA peaks. The combination of this analysis with the in vitro DNA binding preferences of each protein suggests that while FoxH1 preferentially selects the GK and TT sites in the common ChIP-seq peaks, FoxA proteins are distributed between the TT and Forkhead sites.

### FoxH1-GK complexes at atomic resolution

To decipher how FoxH1 selects GK sites and how the extended domain contributes to DNA binding, we determined the crystal structures of FoxH1 proteins from three selected species, *H. sapiens*, *D. rerio*, and *X. laevis*, bound to a 16 bp native sequence belonging to the mouse *Gsc* promoter containing the GG site (Supplementary Table 2). The three GG complexes were refined at 0.98 Å (*D. rerio*, *C*2 space group), 1.47 Å (*H. sapiens*, *P*2$_1$ space group), and 2.8 Å resolution (*X. laevis*, *P*2$_1$2$_1$2$_1$ space group). In each case, the asymmetric crystallographic unit contains a protein bound to a dsDNA

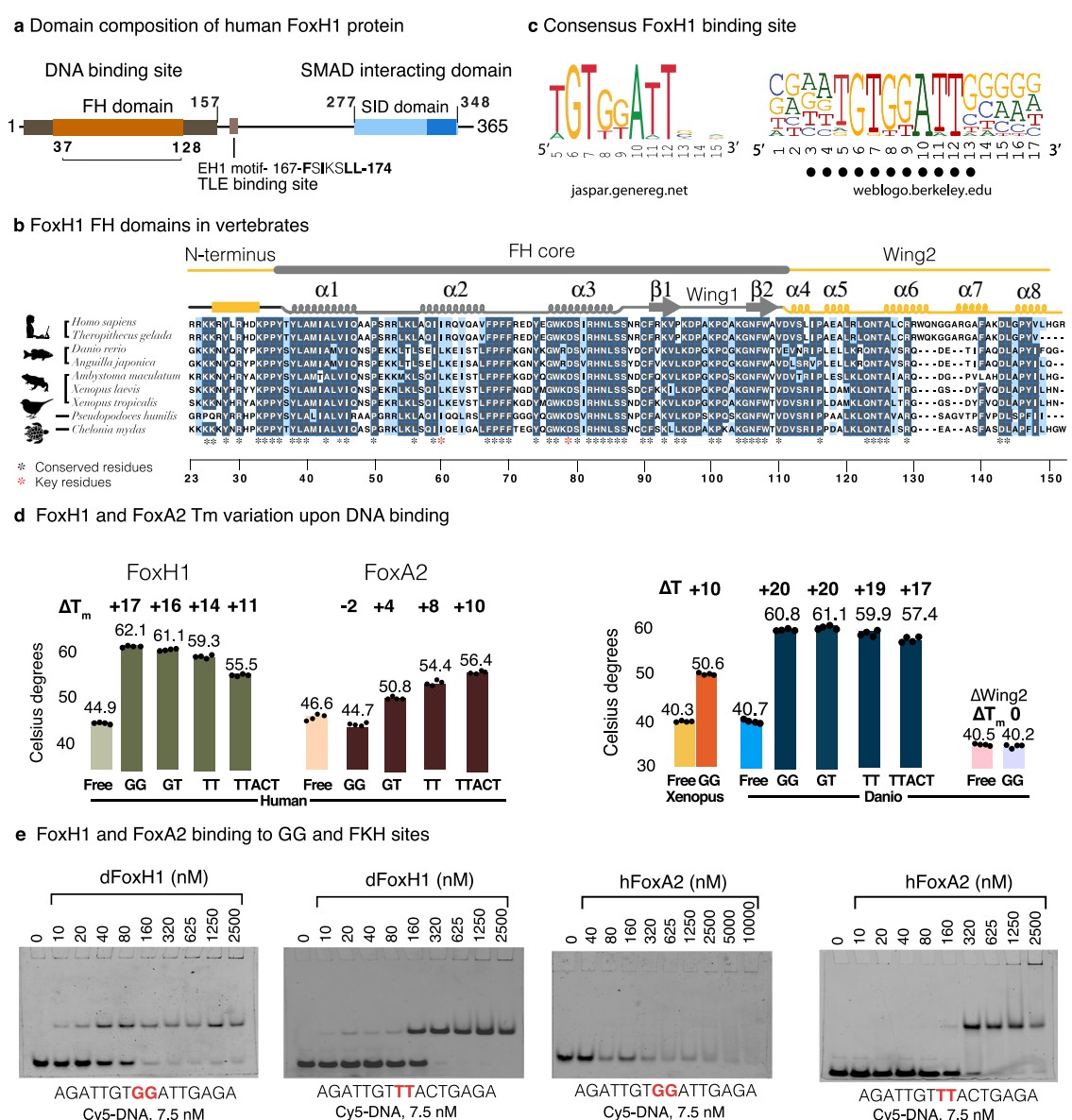

**Fig. 1 | The FoxH1 FH domain binds DNA with high affinity. a** Domain composition of FoxH1 proteins. The FH and SID domains are conserved in vertebrates. **b** Sequence alignment of FoxH1 FH domain in vertebrates. Secondary structure elements observed in the human complex are depicted on the top of the alignment. The FH core domain and the extended N- and C-termini are also indicated. Alignments were generated with MAFFT[76] and the BoxShade server (https://bio.tools/boxshade). Residue conservation is color coded: white: low similarity, blue: high similarity, grey: strictly conserved. **c** Consensus DNA binding. The sequences used to derive this consensus are shown in Supplementary Fig. 1b. Base pairs found to participate in specific protein contacts are indicated. This consensus is almost identical to the JASPAR profile https://jaspar.genereg.net/. **d** Melting temperature

modification of the human FoxH1 and FoxA2 domains in the presence of the different DNAs. The stability of the construct without the C-terminal extension (ΔWing2) is not affected by the presence of DNA. FoxA2 incubation with the GG site induced a decrease in its melting temperature. Melting temperatures correspond to two repetitions and three replicates. Values are summarized in Supplementary Table 1. Source Data are provided as a source data file. **e** Comparison of the binding properties of the FoxH1 and FoxA2 FH domains and two 16 bp cy5-labeled DNA molecules followed by native electrophoretic mobility shift assay (EMSA). The GG motif (in red) is derived from the native *Gsc* sequence. The TT sequence (in red) corresponds to a forkhead site. Whereas FoxH1 and FoxA2 bind well to the forkhead site, binding to the GG site is observed only for FoxH1.

molecule. The root-mean-square deviations (RMSD) between the three vertebrate GG complexes range from 0.47 to 0.65 Å, indicating high structural conservation between the proteins. Although the human construct includes the engrailed homology 1 motif (EH1)[30], this part is not defined in the complex, thereby suggesting that this region does not belong to the globular FH domain. This conclusion is supported by the observation that EH1 motifs are present in several FOX proteins at variable positions in the sequence and not always in the vicinity of the FH domain[34].

Given the overall high similarity of the three vertebrate sequences and the superior diffraction of the zebrafish protein

crystals with GG motifs, we used the zebrafish construct for the structural studies also with the GT motif variant. The GT structure was refined at 1.18 Å resolution (C2 space group). In all complexes, FoxH1 almost covered the 16 bp DNA duplex used for crystallization. The atomic resolution data of the two GK structures, with average B-factors of 19.3 and 17.0 Å[2], respectively, allowed us to determine the position of each atom of the 32 DNA nucleotides, 120 protein residues, and 360/219 water molecules in the GG/GT structures, respectively. The FH fold comprises a three-helix bundle, a double (or triple) β-sheet, and two Wing1 and Wing2 regions. Wing1 corresponds to the loop connecting the two β strands, whereas Wing2 is

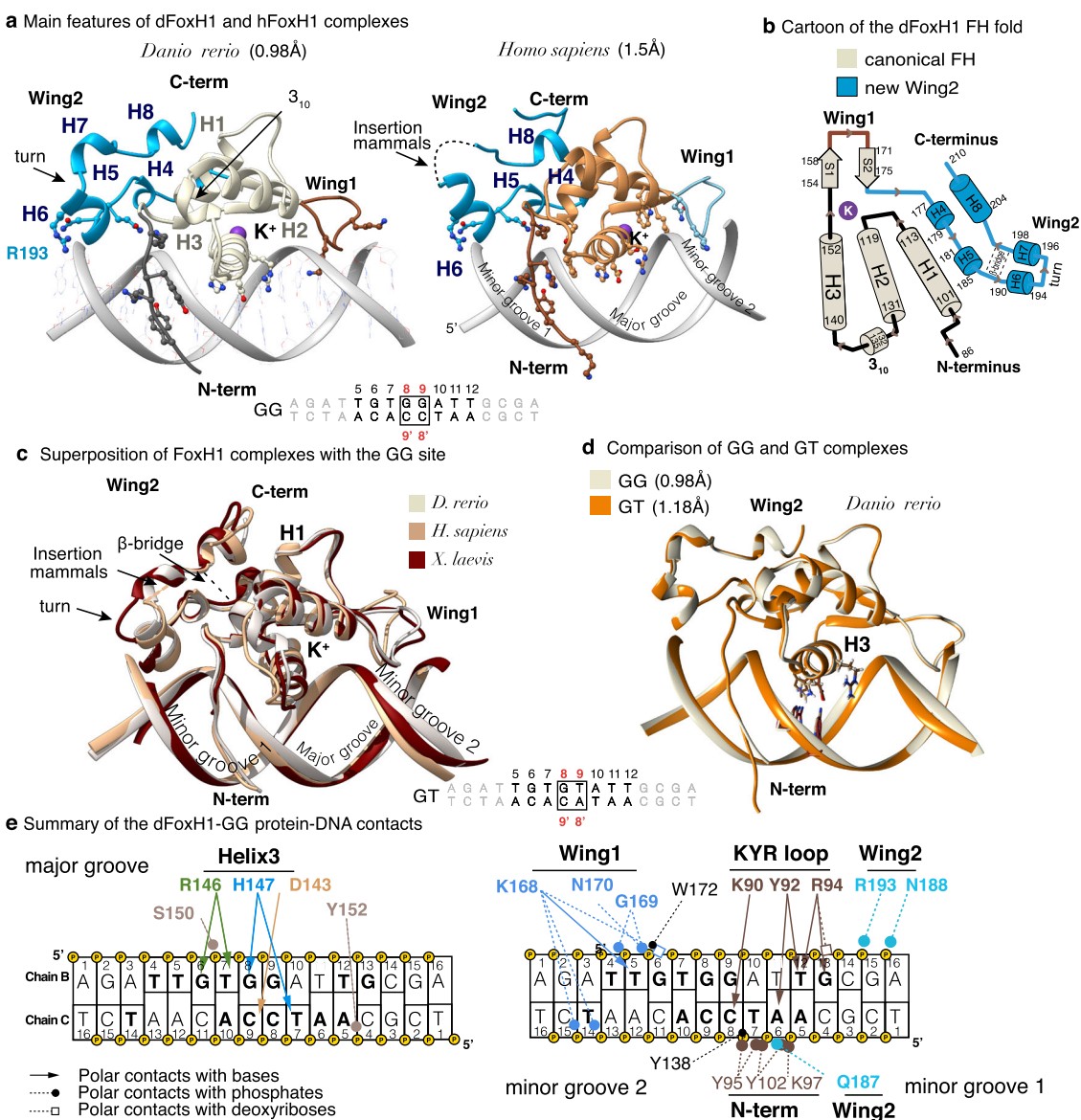

**Fig. 2 | FoxH1-GK complexes structure. a** Zebrafish and human FoxH1-GG complexes. The K$^+$ cation is displayed as a violet sphere. The GG sequence and the elements of secondary structure are indicated. Wing1 and 2 are colored in blue. **b** Schematic drawing of the secondary structure elements. The core domain is shown in beige and the novel Wing2 in blue. **c** Superposition of the human, frog and zebrafish FoxH1-GG complexes. Regions with differences between structures are located after the β strands. As observed in other FH structures[21], the indicated as arrows. **d** Superposition of the zebrafish FoxH1-GG and GT complexes. **e** Specific contacts for the zebrafish complex refined at 0.98 Å resolution showing how FoxH1 recognizes a DNA segment of 15 bp through a rich network of direct contacts involving the major groove and both adjacent minor grooves. Residues are color coded as in panel (**a**).

fold was further stabilized by a K$^+$ cation, which tethered the C-terminus of helix H3 to the beginning of strand S1 (Supplementary Fig. 2a). The cation and its coordination properties were verified using the *CMM server* (https://cmm.minorlab.org/)[35].

It is noteworthy that at such a high resolution, 0.98 Å vs. 1.18 Å, a small difference of only 0.20 Å was associated with a large difference in the number of unique reflections, which doubled in the GG complex with respect to the GT complex (111,470 vs. 54,975 reflections). A summary of the data collection, refinement statistics, and PDB entries is shown in Supplementary Table 5. At the time of this writing, the FoxH1-GG complex at 0.98 Å and Zfp57 protein (PDB:4GZN and 4M9V)[36,37] are the only two TF-DNA complexes solved below 1.00 Å resolution. Regarding other FOX-DNA complexes, only two complexes have been refined below 1.7 Å, namely FoxG1 and FoxN1 (PDB:7CBY and 6EL8)[21].

### FoxH1 has a specific FH fold

FoxH1 adopts the canonical FH domain core for the conserved part of the domain. However, in this case, the core domain is flanked by two additional regions that are well-defined in the electron density maps: the N-terminal loop and an unusually long Wing2 (50 residues in human FoxH1 and 45 in zebrafish) (Fig. 2a, b). The core domain comprises a three-helical bundle (H1-H2-H3), followed by a pair of anti-parallel β strands (S1, S2) (Fig. 2b). In the FoxH1 complexes bound to the GG site, Wing1 and Wing2 are well-ordered and, together with the N-terminal loop, they contribute to specific recognition of DNA. In the three FoxH1 complexes, Wing2 folds as a series of short α-helical turns. These helical turns enable a network of interactions with the domain core and with the N-terminal loop, thereby defining a novel FH domain fold (Fig. 2c and Supplementary Fig. 2b, c). The superposition of the GK complexes (Fig. 2d) confirms the high structural similarity between them, with the protein being able to interact with both GG and GT

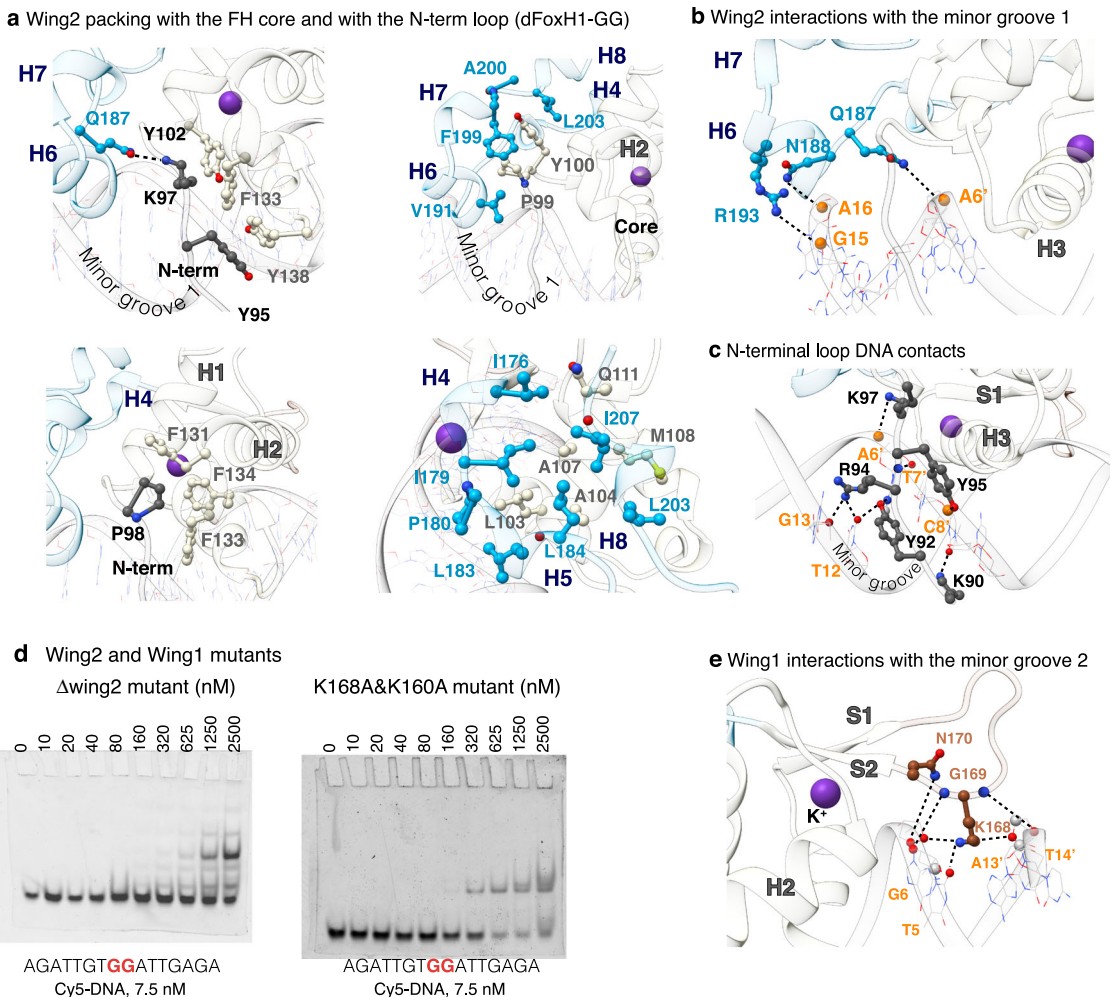

**Fig. 3 | Minor groove interactions. a** Four views of Wing2 and the N-terminal loop displaying side chain–side chain contacts contributing to the distinctive FoxH1 FH fold. Residues are color coded as in Fig. 2a. **b** Wing2 binding to minor groove 1. HBs are indicated with dashed lines. Protein residues, nucleotides and phosphates involved in the interaction are shown and labeled. **c** Specific interactions between the N-term region and the DNA minor groove 1. **d** Effect of point mutations and deletions of residues at both Wing regions on DNA binding. **e** Specific interactions between the Wing1 region and the DNA minor groove 2.

motifs efficiently (RMSD of 0.23 Å). A summary of the zebrafish/human protein-DNA interactions for the GG complexes is provided in Fig. 2e and in Supplementary Fig. 2c, respectively.

Several differences located in Wing2 and also in the loop preceding H3 reflect the amino acid variations between the three species. Wing2 starts with two $3_{10}$ helices (H4 and H5, Fig. 2a, b), followed by an isolated β-bridge formed by Lys185 and Asp202 in zebrafish and by Arg121 and Asp143 in humans (Supplementary Fig. 2d). This β-bridge connects two antiparallel-oriented regions and stabilizes the turn connecting H6 and H7 helices. The β-bridge is followed by the helix H8, which is a $3_{10}$ helix in zebrafish. In the human and frog sequences, which are longer at the C-terminus, we detected a short coil fragment that turns back and runs antiparallel to H8. A comparison of the three GG complexes revealed the presence of a mammalian-specific insertion of five amino acids, 134-GlyGlyAlaArgGly-138 (Figs. 1b, 2c), which is located within H7. This insertion cannot be accurately traced in the human structure due to intrinsic flexibility. We also observed that a hydrogen bond (HB) between Asn187 and Lys97 (the latter located at the N-terminus) triggers numerous interactions between Wing2 and the FH core. For instance, Arg178 (in H4) interacts with the turn between H3 and S1, while Leu183 (in H5) binds the last residue of the N-terminal loop (Ser 101) whereas Leu203, Tyr206, Ile207, and Phe208 (in H8) interact with residues in both H1 and H2. Some of these interactions are depicted in Fig. 3a.

## The two Wing regions and the N-terminal loop contribute to the recognition of minor groove 1

The Wing2 region also interacts with minor groove 1, directly with the backbone (Gln187 to A6′, Asn188 to A16, and Arg193 to G15) and indirectly by guiding the N-terminal loop also towards this groove, where it establishes several base-specific HBs (Fig. 3b, c), for instance, between Tyr92 and A6′, and between Arg94 and T12 and G13. These interactions are well-defined, as depicted in the snapshot showing the atomic-resolution 2Fo-Fc electron density maps of the zebrafish GG complex contoured at 1.0 sigma (Supplementary Fig. 3a). In addition, in the GT complex, Lys90 of the KYR motif enters the major groove to form a HB with A8′ of the GT pair (T9-A8′ bp). This network of interactions explains our observation that the DNA binding properties of the mutant are affected in constructs lacking the Wing2 region (Fig. 3d). Point mutations of two conserved aromatic residues that contribute to Wing2 packing promote protein aggregation during purification, probably because the protein is partially unfolded.

The remaining interactions with the minor grooves arise from Wing1, which contacts minor groove 2. These contacts involve a pair of specific HBs from Lys168 to T5 and T14′, as well as with the ribose of C15′ (Fig. 3e, Supplementary Fig. 3b) in the GG complexes, and from Lys 168 and Lys160 to C15′ in the GT complex (Supplementary Fig. 3c). In fact, mutations of these Lys residues to Ala decrease affinity for DNA (Fig. 3d).

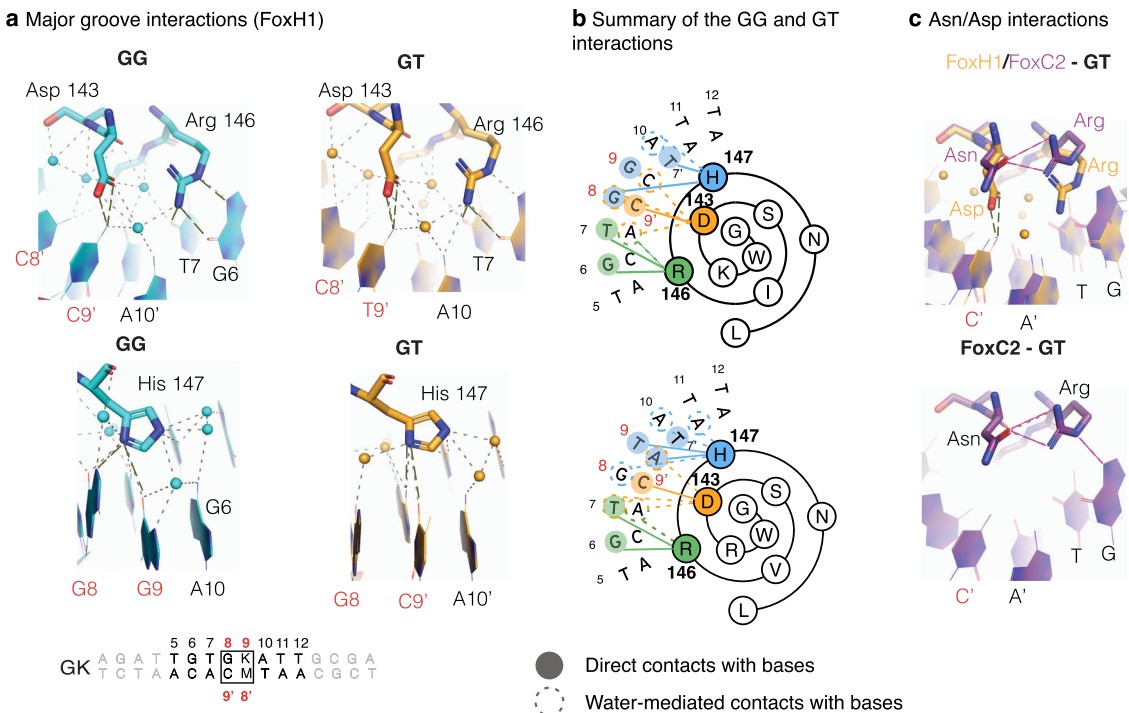

**Fig. 4 | Major groove interactions. a** Snapshots highlighting direct and water-mediated polar contacts from the Asp143, Arg146 and His147, located in H3, with nucleobases (distance up to 3.6 Å) of the major groove. Contacts of Asp and Arg with the DNA in the GG and GT complexes are similar. The main differences concentrated at the His residue, which binds to the G or T sites. In the GG complex, the Asp residue is found in double conformations, with the main conformation (75% occupancy) corresponding to that seen in the GT complex. In the other orientation, the side chain is rotated approximately 90 degrees. For clarity, we show only the major Asp and DNA conformation. Electron density maps are provided in Supplementary Fig. 4. **b** Summary of the FoxH1 interactions with GG and GT sites represented as a 2D Wenxiang diagram. **c** Comparison of the direct interactions observed in the FoxH1 GG and GT complexes and the FoxC2-GT complex. Whereas in FoxH1 both Asp and Arg side chains participate in a direct HB with the GK site, FoxC2 displays interactions between the equivalent residues and a polar contact from the Arg residue with the G site.

These direct and specific interactions with DNA from the two Wing regions and the N-terminal loop are rare in other FOX proteins and, when observed, they usually occur with the backbone DNA in a non-specific manner.

### Key roles of helix H3 Asp and Arg residues in the recognition of the GK site at the major groove

In all FOX proteins, helix H3 is docked into the major groove, where it interacts with the TGTTTA region of the forkhead motif. Specific binding to the major groove is established through HBs using three conserved residues in the helix: Asn (position i), Arg (i + 3) and His (i + 4). FoxH1, which binds to TGTGKATT sites, also has His and Arg residues conserved at positions (i + 3) and (i + 4), but it has an Asp residue instead of Asn at position i. The Asn to Asp substitution is unique to FoxH1. In fact, 47 out of the 49 human FOX proteins contain a conserved Asn residue (Supplementary Fig. 1a). In FoxH1, Arg146 interacts with the TGT region through a bidentate HB interaction with G6 (O6 and N7 atoms), and HB and van der Waals contacts with T7 (O6 atom and its methyl group, respectively). The His and Asp side chains bind to the GK site. The His residue recognizes both position 8 and 9 via direct HBs from ND1 to G8 (O6) and to C8' (N4) in the GG complex or by a direct HB with the T9-A8' bp and a short-distance water-mediated HB to G8 (O6) in the GT complex (Fig. 4a, b).

The structures indicate that the Asp in FoxH1 allows the formation of a HB with the cytosine base of the G8-C9' pair. Although there is a complex of a FOX protein bound to a GK site in the literature (FoxC2, PDB 6akp), in this case, the conserved Asn residue cannot establish an efficient interaction with the G-C bp as its side chain points away from the DNA (Fig. 4c). This feature explains why FoxC2 has a lower affinity for GT than for TT sites, where the Asn residue actively participates in DNA binding[38]. A similar absence of interaction between Asn and the G-C pair could account for the DNA binding preferences of FoxA2 (Fig. 1e). These contacts indicate that the switch of Asn to Asp in FoxH1 is key for efficient interaction with the G-C bp.

### Binding to canonical forkhead motifs

Since FoxH1 interacted with TT sites in binding assays and in ChIP-seq data, we also determined the corresponding complexes for comparison. In these complexes, we used either a GG to TT variant of the native *Gsc* sequence (TGTTTATT) or the forkhead motif (TGTTTAC) (Fig. 5a). These complexes were refined at 2.2 Å and 2.1 Å resolution, respectively (Supplementary Table 1). The superposition of these two TT complexes with the GG one (RMSD values of 0.70 and 0.74 Å, respectively) revealed that the main differences concentrate at Wing2, which seems to be less ordered in the TT and forkhead complexes (indicated with an asterisk in Fig. 5a). We also observed a slight modification of the Wing1 orientation in the forkhead complex, in the region after S1, with respect to the GG and TT complexes.

Apart from these small differences, in all complexes, FoxH1 interacts similarly with the common part of the DNA, whereas the interactions with the His and the Asp residues are adjusted to recognize the new TT pair. For instance, the Asp residue establishes HBs with the A9' bases of the T8-A9' pair (Fig. 5b). With respect to the variations introduced by the T → C modification at position 11 in the forkhead DNA, FoxH1 tolerates certain modifications at the DNA sequence of the minor groove. This position is recognized by the aromatic ring of Tyr92 located at the N-terminus and, in both complexes, Tyr92 establishes similar interactions with either the A or with the G nucleotides in the complementary DNA chains (Fig. 5c).

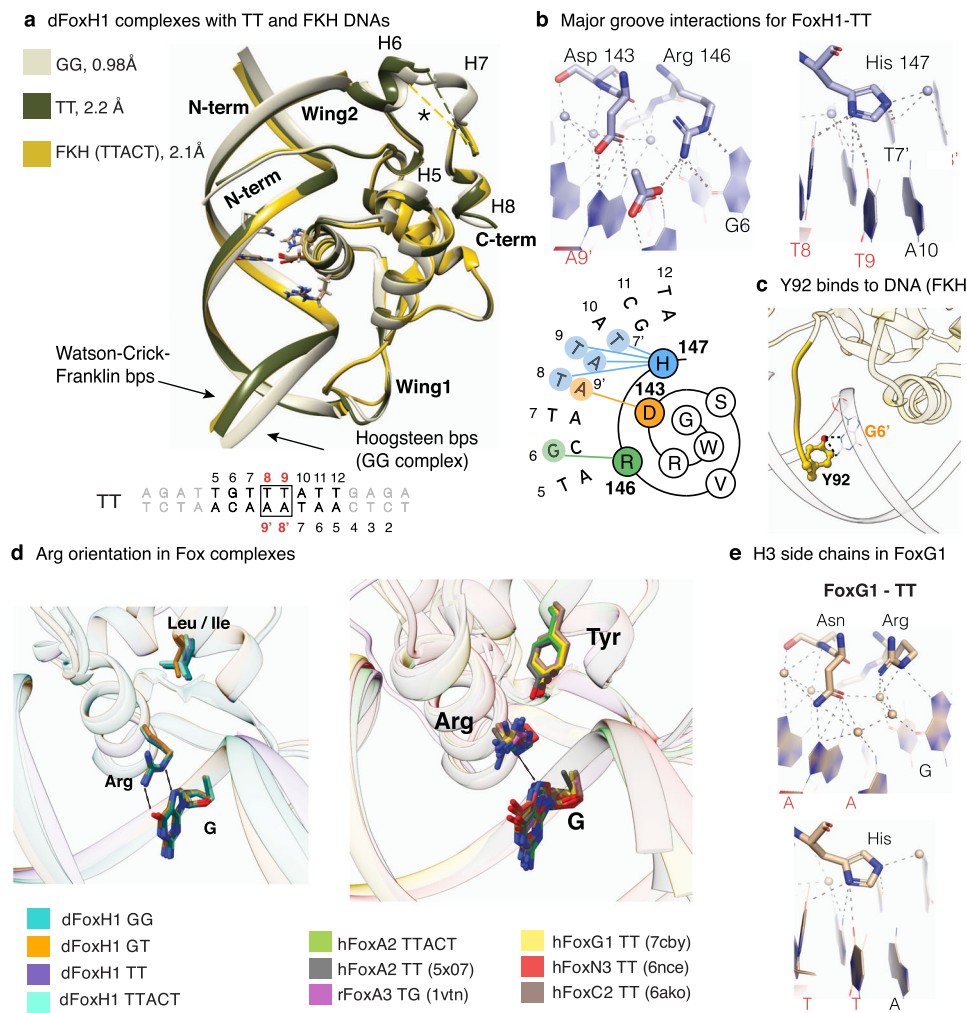

**Fig. 5 | Binding to canonical forkhead motifs. a** Superposition of the three zebrafish FoxH1-DNA complexes with the GG (beige) and two TT sites (dark green and gold). **b** Snapshots highlighting direct and water-mediated polar contacts from the Asp143, Arg146, and His147, located in H3, with nucleobases (distance up to 3.6 Å) of the major groove for the FoxH1-TT complex, with a summary of the FoxH1 interactions with the TT site represented as a 2D Wenxiang diagram. **c** Tyr92 specific interaction with the minor groove in the FHK DNA. **d** Different

rotamers of the Arg residue in FoxH1 and in six other FOX complexes (PDB entries indicated). In FoxH1, the Arg residue is close to the DNA and participates in direct contacts, whereas in the remaining complexes the side chain is rotated away. **e** FoxG1-TT complex (PDB:7CBY). Snapshots highlighting direct and water-mediated polar contacts from the Asp, Arg and His residues (located in the H3 helix), with nucleobases (distances up to 3.6 Å).

## Similarities and differences between FoxH1 and other FOX-DNA complexes

In addition to the essential differences observed in the DNA interactions caused by the Asp/Asn modification, the strong bidentate Arg-Gua interactions present in the FoxH1 complexes with the common TGT motif are absent in all the other FOX-forkhead motif complex structures determined to date[21,32,38–42] and the pair of FoxA2 structures bound to the TT sites, which we also determined (Fig. 5d and Supplementary Table 5). This difference is caused by the presence of a conserved Tyr residue in H2, which is substituted by a Leu/Ile amino acid in FoxH1. In the presence of the Tyr residue, the Arg side chain is partially sequestered by interaction with the hydroxyl group of the Tyr residue, thereby preventing the formation of direct bidentate bonds with the G base (Fig. 5d). In the most favorable cases, as in the FoxG1 complex (Fig. 5e), this interaction is water-mediated. The presence of Leu/Ile in FoxH1 (instead of the aromatic Tyr) provides rotational freedom to the Arg residue, thereby facilitating the interaction with G6 and increasing DNA affinity and specificity.

Overall, the comparison of FoxH1 to other FOX complexes reveals how two specific sequence modifications, namely Asp vs. Asn in H3 and Leu/Ile vs. Tyr in H2, allow FoxH1 to directly recognize the GG and GT

bases without losing its capacity to interact with the TT sites characteristic of all FOX proteins.

## DNA shape analysis

To explore the impact of the protein-DNA interactions on the DNA topology, we compared the DNA shapes of the FoxH1 complexes with other FOX structures available in the PDB and with the pair of complexes we determined here for FoxA2 using Curves+ [43] (Supplementary Fig. 5d). Small differences were detected at the major groove, whereas the minor grooves were found to be slightly wider in the FoxH1 complexes than in other FOX structures. These differences at the minor groove of FoxH1 are probably caused by the abundant number of protein-DNA interactions present in this protein, which are absent in other members of the FOX family (Supplementary Table 3). We also detected the presence of Hoogsteen base pairs (HGs) in the GG complex (indicated with an arrow in Fig. 5a and shown in detail in Supplementary Fig. 5e), instead of the Watson−Crick−Franklin (WCF) base pairing observed in other structures. HGs require flipping of the purine base by 180° with respect to the corresponding WCF base pairs and the protonation of the Cytosine N3. Although Hoogsteen base pairs are still infrequent in structures, their proper identification have gained

**Fig. 6 | FoxH1 binding to reconstituted mononucleosomes. a** Schematic representation of the 601 147-NCP WT and variants used in this work. The WT sequence is shown on the top and the modifications introduced in the variants are highlighted. Changes in the m2 sequence were introduced to remove FoxH1 pseudo-motifs. A–D variants represent the different positions of the FoxH1 specific motif explored in this work. A model of a nucleosome (based on PDB:5oxv) with FoxH1 bound to the A-D variants is shown in three different views. **b** Titration of the Widom601-FoxH1 NCP-147 (+6) with the FoxH1 FH domain followed by EMSAs. Arrows indicate the different species. In violet, species present in control lanes. In green, NCP-protein complex. **c** Quantification of the affinities between the FoxH1 FH domain and either free 601 variants or nucleosomes, as shown by Biolayer Interferometry. Bars represent fitted $K_D$ values and error bars represent standard error of the fit ($n = 5$). Source Data are provided.

attention in the last decade[44] to clarify their functional role as they are found near mismatches or DNA lesions[45].

## FoxH1 binding to reconstituted mononucleosomes

The DNA-binding domain of FOX proteins has structural similarity to the folded domain present in the linker histones H1 and H5 as they all share the same winged-helix domain fold[40,46,47]. This structural convergence, originally identified in FoxA3 with H1, is thought to allow pioneer factors to gain access to chromatin, thus facilitating the subsequent binding of other transcription factors and chromatin remodeling proteins[7]. Although FoxH1 has been reported as a pioneer factor[6,8], how FoxH1 binds to nucleosomes is currently unknown. To address this question, we first used the Widom 601 sequence and a compact 147 bp nucleosome core particle (NCP)[48]. We observed interactions between FoxH1 and this NCP using electrophoretic

mobility shift assays (EMSA), although no clear band of the complex was observed, probably due to interactions with FoxH1 pseudo-motifs present in the sequence. When these sites were mutated, non-specific binding was undetected (Fig. 6a, Supplementary Fig. 6a). As a control for sequence-agnostic binding, we used the linker histone H1A and the same 147 bp Widom 601 nucleosome, since this histone should recognize the linker and not the compact NCP. To enhance the specificity, we introduced four bp modifications to adapt a potential non-specific site present in 601 to the TGTGGATT motif (Variant A, Fig. 6a, b). In this case, we observed specific interactions between the protein and the compact mononucleosome, as inferred by the discrete band observed in the EMSA experiment at protein concentrations as low as 47 nM. To clarify whether the positioning of the FoxH1 specific motif conditioned the binding, we prepared three more NCP variants with the FoxH1 motif situated at superhelical locations (SHLs) −6, +2.5, and

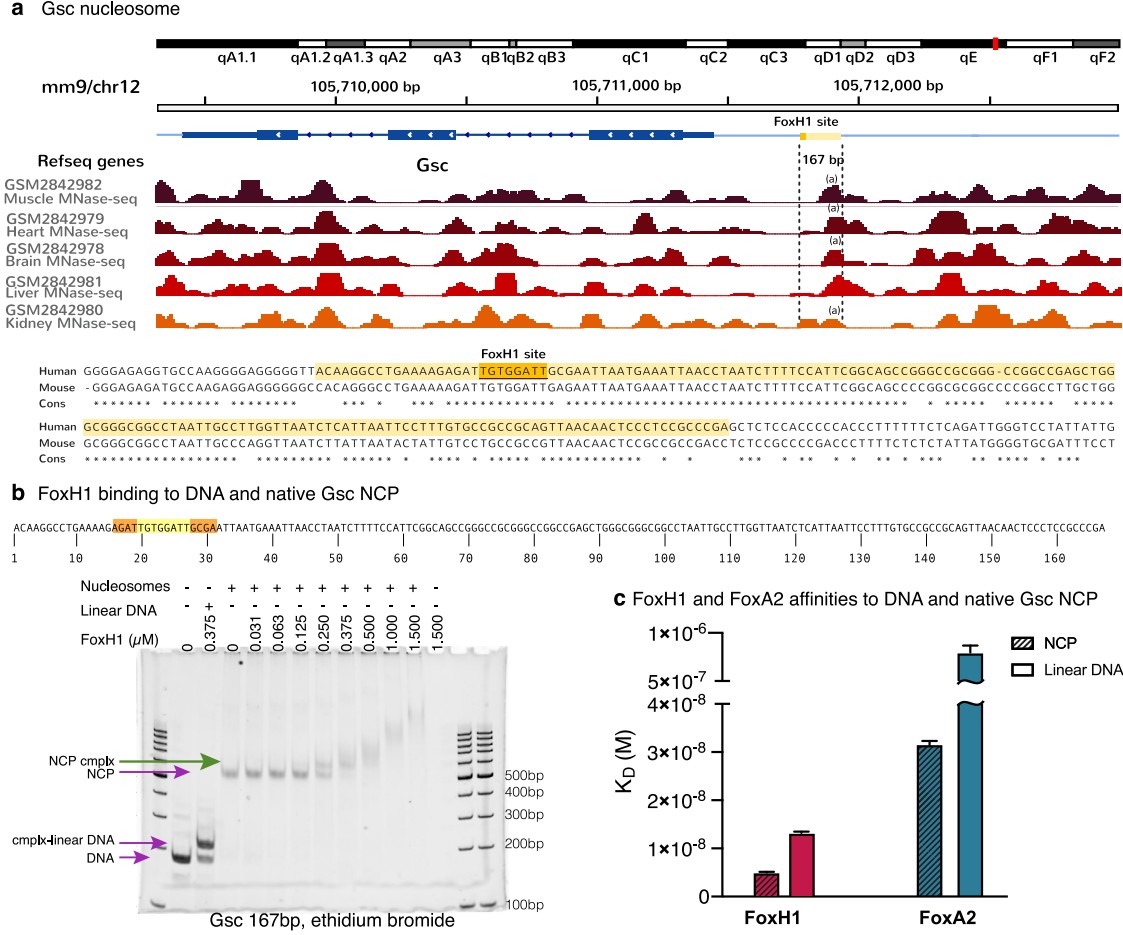

**Fig. 7 | interactions between the FoxH1 FH domain and a native *Gsc* promoter nucleosome. a** *Gsc* boundaries (dashed lines) were selected based on the MNase seq information described in GSM2842982 (mouse). The boundaries are more similar in muscle, heart, brain and liver than in kidney datasets, with differences perhaps indicating NCP dynamics in the different conditions. In this work, we used the *Gsc* sequence corresponding to the (**a**) site. The *Gsc* human NCP sequence (in light yellow) contains the FoxH1 site (highlighted in orange). Both human and mouse sequences are highly conserved. **b** Titration of the native *Gsc*167 with FoxH1 FH domain. The *Gsc*167 sequence and the FoxH1 motif are indicated. DNA ladder is included as a reference for the linear DNA. **c** Comparison of the binding affinities of FoxH1 and FoxA2 to the Gsc sequence (NCP and linear DNA). Bars represent fitted $K_D$ values and error bars represent standard error of the fit ($n = 5$). Source Data are provided.

+0.5 (Variants B to D, Fig. 6a). To measure the interaction of the different variants and linear DNAs, we used Biolayer Interferometry (BLI) (Fig. 6c). Based on the measured affinity values (in the nanomolar range), the FoxH1 motif can be located even at positions close to the nucleosome center (+0.5) while retaining a strong interaction with FoxH1. Of the four constructs tested, FoxH1 showed a preference for binding nucleosomes over linear DNA when the motif is located at SHL +6, +2.5, and +0.5 (Fig. 6a).

We next sought to characterize the interactions between the FoxH1 FH domain and a native *Gsc* promoter nucleosome that has the 16 bp sequence used in the complexes. To identify the nucleosome boundaries, we selected various published MNase-seq experiments performed on adult mouse tissues (brain, kidney, muscle, liver and heart) (Fig. 7a). We reasoned that in these tissues the nucleosome should be intact since FoxH1 and Goosecoid are not expressed and might resemble the nucleosome definition prior to FoxH1 binding to this promoter during embryonic development. These datasets showed good agreement between the 167-*Gsc* nucleosome boundaries despite the difference in conditions studied. To test the stability of the native nucleosome, we performed an enzymatic digestion under standard conditions (37 °C for 30 min) using three enzymes (StuI, AscI and BbvI) that specifically cut the linear DNA sequence at different sites (Supplementary Fig. 6b). When the 167 *Gsc* nucleosome was treated with

these enzymes, only StuI (whose restriction site is located at the 5′ end) is able to cut it. This result indicates that the StuI site is exposed in the nucleosome, whereas the other sites are protected. Based on the digestion pattern and assuming the most extreme case, where the nucleosome is formed using the remaining bps and shifted towards the 3′ end, the FoxH1 site would be still located in the supercoiled area of the NCP (modeled in Supplementary Fig. 6b), although its precise location might vary slightly due to the natural breathing of the NCP. With this nucleosome, we analyzed its binding properties by native electrophoresis and observed a defined shift of the nucleosome in the presence of FoxH1 (Fig. 7b). We quantified the affinity of protein-DNA interactions in Biolayer Interferometry (BLI) assays, which, as with the model 147 FoxH1-601 NCPs, are in the nanomolar range (Supplementary Fig. 6c). Since at the end of the *Gsc* sequence there is a TGTTAAC motif that almost fits the so-called 'degenerate FoxA motif on nucleosomes', we also measured the interactions of FoxA2 with the same DNA molecules for comparison[49]. We observed that the FoxH1 and FoxA2 FH domains interact with higher affinity to the mononucleosomes than to linear DNA, although the affinity is always higher for FoxH1 (NCP: 5 nM and 30 nM, Linear DNA: 13 nM and 800 nM, Fig. 7c). The results of mononucleosome binding corroborate the role of FoxH1 and FoxA2 proteins as pioneer factors as they show a preference for binding to mononucleosomes containing either the

canonical FoxH1 site or a degenerate FoxA motif over binding to linear DNA. These observations align with similar data obtained for other TFs, such as Oct4 and Sox2[50,51].

We also used this binding assay to quantify the effect of point mutations in the FoxH1 FH domain located at the N-terminus. The functional role of residues located here was postulated after genetic analyses in zebrafish embryos (sur/schmalspur)[5,52] and more recently has been highlighted by mutations detected in human tumors (https://cancer.sanger.ac.uk)[53]. The mutation of the Arg and Lys residues in zebrafish (R94H and K97N mutants) induces severe developmental pathologies. As observed in the $K_D$ measurements, all mutations show reduced DNA-binding capacity with respect to the WT construct using the native *Gsc*-NCP (Supplementary Fig. 6d).

## Discussion

### FoxH1 has a unique fold and DNA binding ability

Our structural studies reveal three essential features of the FoxH1 FH domain that enable DNA recognition through a network of interactions with the major and minor grooves.

First, FoxH1 has an extended fold that almost duplicates the length of the FH domain in other FOX proteins. Given the high reliability of AlphaFold2 predictions for folded domains, we downloaded models of all human FOX proteins from the AlphaFold protein structure database, including FoxH1. We analyzed the Wing2 region in these models for the presence of extended FH domains in other FOX proteins[54,55]. Although AlphaFold2 predicted several helical segments for FoxH1 and also for FoxN2 and FoxN4, we were unable to identify structures like the one present in FoxH1 for other FOX proteins. In FoxN proteins, the predicted helices are oriented differently from FoxH1 and do not appear to participate in interactions with DNA.

Second, FoxH1 also has two specific amino acid modifications, Asp versus Asn and Leu/Ile versus Tyr, which are essential for major groove recognition and binding to GK sites. However, whereas the substitution of Tyr for Leu/Ile has an allosteric effect on DNA binding, the role of the Asp residue is direct. Acidic residues, such as aspartate and glutamate, are not very common at the protein-DNA interfaces of TFs, probably due to electrostatic repulsions with the phosphate groups of the DNA. Of note, under physiological pH (values ranging from pH 5 to pH 7.5), the Asp residue appears as the negatively charged aspartate form and is able to establish HBs with cytosine and adenine bases, as observed in FoxH1 complexes and as also reported in the literature for other TFs[56,57]. In contrast, the Asn present in other FOX proteins has a polar side chain and selects only the adenine bases in complexes containing TT sites. This scenario is exemplified by FoxC2 structures with TT and GT motifs (Fig. 4b).

Third, the specific interactions between FoxH1 and DNA seem to extend from the initially proposed canonical motif (positions 5–12 of our DNA sequence) to include specific interactions with three additional base pairs. At the 5′ site, these additional interactions occur through contacts from Lys104 (human sequence) or Lys168 (zebrafish) located in Wing1 with positions 2, 3, and 5 of DNA. The additional contacts at the 3′ site arise from interactions of the N-terminal loop to G13. The conservation of the protein residues involved in these additional interactions and the preference for G-C bases at position 13 underscore the role of these contacts in enhancing the affinity of FoxH1 interactions with *cis*-regulatory elements. However, since FoxH1 binds to various loci, it probably tolerates different bps at these additional positions.

These three FH domain features distinguish FoxH1 from all other FOX proteins, both in terms of structure and DNA binding properties. The specific structural properties of FoxH1 have the potential to facilitate the design of selective small-molecule compounds to expand our understanding of the roles of FoxH1 in development and in reprogramming. Given the distinct fold and DNA sequence preferences shown by FoxH1, this knowledge might also assist the

development of new therapeutic strategies to target increased expression of this protein in acute myeloid leukemia[27].

### FoxH1 role as a pioneer factor

In accordance with the documented role of FoxH1 as a pioneer factor, we confirmed that the FH domain of FoxH1 binds with nanomolar affinity in vitro to Widom 601 147-NCPs containing the FoxH1 site at several positions of the sequence and also to a native *Gsc* 167-mononucleosome. Interestingly, FoxH1 displays a preference for nucleosomal DNA over linear DNA in four of these binding assays, when the FoxH1 site is located at +6, at +2.5, and at +0.5 in model 601-NCPs and also with the native Gsc mononucleosome (Fig. 7b, c). All these areas are exposed to the surface of the mononucleosome and fully accessible in our models. In fact, the presence of the extra elements of the tertiary structure do not affect the shape complementarity between the protein and DNA, even when the DNA motif is wrapped around a histone octamer. Thus, the structural features of FoxH1 are compatible with its proposed pioneer factor function, even despite (or favored by) its large DNA-binding surface area.

### Functional implications of GK recognition by FoxH1

It has been proposed that FoxH1 and FoxA proteins participate in a molecular "hand-off" mechanism to maintain enhancer activation of specific cis-regulatory elements during the maternal-to-zygotic transition in Xenopus[8]. The structural information we gathered for the DNA binding specificities of FoxA2 and FoxH1 FH domains is compatible with the hypothesis whereby FoxA proteins substitutes for FoxH1 preferentially at TT sites, with FoxH1 remaining bound to GK sites even when the concentration of this protein is low. The presence of GK sites in many FoxH1 targets may have been optimized, together with the key differences in the FoxH1 structure, thus avoiding the misregulation of essential genes that contain GK sites by other FOX proteins optimized to select TT motifs.

## Methods

### Protein expression and purification

Wild-type FoxH1 (*Danio rerio*: Uniprot Q9I9E1, aa 86–210 and 86–175 (Δwing2), *Homo sapiens*: Uniprot P75593, aa 1–185, *Xenopus laevis*: Uniprot P70056, aa 97–236) and FoxA2 protein (*Homo sapiens*, Uniprot: Q9Y261, aa 153–273) were cloned in pOPINS using synthesized DNA templates with optimized codons for bacterial expression (Thermo Fisher Scientific). Point mutations were produced by site-directed mutagenesis PCR reactions and confirmed by DNA sequencing (GATC Biotech). All proteins were expressed fused to an N-terminal His-tag SUMO-tag followed by the Ulp1 peptidase cleavage site in *E. coli* B834(DE3) strain essentially as described[6,23,58]. Cells were grown at 37 °C in Terrific Broth and induced with IPTG (0.5 mM) at an OD600 of 0.8. After overnight expression at 20 °C, bacterial cultures were centrifuged and cells were lysed at 4 °C (EmulsiFlex-C5, Avestin) in 50 mM Tris, 400 mM NaCl, 40 mM imidazole, 1 mM TCEP and Tween 20 0.2% V/V pH 8 at 25 °C in the presence of lysozyme and DNase I. Supernatants containing the soluble proteins were diluted until a conductivity of 10 mS/cm was reached, then loaded into HiTrap™ SP HP 5 mL column and eluted by a NaCl gradient to remove non-specifically bound bacterial DNA, using a NGC™ Quest 10 Plus Chromatography System (Bio-Rad). Fractions containing the protein of interest were pooled, dialyzed to reduce the NaCl concentration and cleaved with recombinant Ulp1 (SUMO protease) overnight at 4 °C. Cleaved proteins were loaded into a HiTrap™ SP HP 5 mL column (GE Healthcare Life Science) to separate the SUMO tag from the FoxH1/FoxA2 FH domains. Finally, all FH domains were purified by size exclusion chromatography on a HiLoad™ 16/600 Superdex™ 75 pg (GE Healthcare Life Science) in 50 mM Tris, 150 mM NaCl and 1 TCEP at pH 7.2 at 25 °C (buffer A) and kept at −80 °C.

## Cell maintenance

HEK293T cells were cultured DMEM medium supplemented with 10% FBS, 1% penicillin/streptomycin and 4 mM L-glutamine. Cells were maintained at 37 °C and 5% CO2.

## Protein production in HEK293T cells

To prepare cell lysates, HEK 293T cells were transfected with full-length Flag-FoxH1 (Flag-FoxH1 was a gift from Stefan Koch, Addgene plasmid #153125)[59]. Cells expressing Flag-FoxH1 fusion construct were harvested 48 h post-transfection. Cells were washed twice in cold phosphate-buffered saline (PBS) and lysed in 200 μL of RIPA buffer (50 mM TRIS pH 7.4, 150 mM NaCl, 1 % NP-40 and protease inhibitors). Lysed cells were incubated for 15 min on ice and then centrifuged at maximum speed for 15 min at 4 °C. Supernatants were collected and total protein concentrations were determined using DC™ Protein Assay Kit II (BioRad). We also immunoprecipitated (IP) the FL-FoxH1 protein using anti-FLAG®M2 Affinity Gel (Sigma) and eluted by competition with 3X FLAG® Peptide (Sigma). Purified protein was then concentrated with Amicon Ultra-0.5 Centrifugal Filter 10 kDa (Merck Millipore). Its folding and functional properties were tested by EMSA, using a fixed amount of Cy5-labeled GG dsDNA (0.7 ng) was incubated with sequential dilutions of the purified protein (Supplementary Fig. 1d).

## Motif enrichment

We downloaded ChIP-Seq data (bed format) from the Gene Expression Omnibus (GEO) Database, with accession numbers GSM2263597 (FoxA proteins, stage 10.5), GSM2263590 (FoxH1, stage 8), GSM2263591 (FoxH1, stage 9) and the reanalysis of the data from GSE53652[22] available at GEO series GSE85273[8]. We then analyzed the presence of FoxH1 motifs (TGTGKATT and TGTTKAT sites) and the FoxA forkhead motif (TRTTTAC) in the form of enrichment with respect to reshuffled primary sequences using SEA 5.4.1[33] in each of the datasets. Fasta files for the analysis were generated from the bed files and *X. tropicalis* 7.1 genome (http://www.xenbase.org/, RRID:SCR_003280) using BED-Tools 2.24[60]. We used STREME 5.4.1[61], a motif discovery program, to ensure that the FoxA motif, as defined in JASPAR 2022[62], was the most enriched motif in these peaks. In fact, we obtained this motif ranked with a p-value of 1.6e-42. Default options were used for both SEA 5.4.1 and STREME 5.4.1 programs.

## Nucleosome preparation and binding assays

**DNA preparation.** 147 bp Widom DNA fragments with and without the FoxH1 binding site and the 167 bp fragment from the native Gsc promoter, containing the FoxH1 binding site were amplified by polymerase chain reaction. A 40X reaction was prepared using Phusion polymerase (Thermo Scientific) following the manufacturer's instructions, in the presence of 5′-biotinylated, 5′-Cy5 or non-modified forward primers and non-modified 3′ primers. Final DNA sequences are included in Fig. 6.

Amplicon was purified using the QIAquick PCR Purification kit from Qiagen and eluted in water. Eluent was then pooled, lyophilized, and resuspended to a final concentration of approximately 1.5 mg/mL.

**Human core histone preparation.** Each of the four canonical human core histones (H2A, H2B, H3.2, and H4) in a pET3 vector was independently transformed into *E. coli* BL21 (DE3) cells and grown at 37 °C until reaching an OD600 of 0.6–0.8, at which time isopropyl-β-d-thiogalactopyranoside (IPTG) was added to a final concentration of 0.5 mM. Cultures were grown for another 3–4 h before harvesting by centrifugation at 5000 × g for 10 min and freezing at −20 °C.

For all histones, the pellet from a 1 L expression culture was thawed and resuspended in a lysis buffer containing 1 X PBS and 1 mM DTT. The sample was lysed by rod sonication with a Branson digital sonifier (40% amplitude, 5 s on, 10 s off, 90 s on-time). Lysate was then

clarified by centrifugation at 20,000 × g for 25 min. Supernatant was discarded, and the insoluble pellet was then resuspended in a buffer containing 6 M guanidine hydrochloride, 1 X PBS, and 1 mM DTT. Inclusion bodies were extracted from the resuspended pellet by rotating overnight at 4 °C. Samples were then clarified as before by centrifugation, and the supernatant was then passed through a 0.22 μm filter. Next, core histones were purified by RP-HPLC on a preparative scale C-18 column. HPLC buffer A was 0.1% (v/v) TFA in water, and HPLC buffer B was 90% (v/v) acetonitrile and 0.1% (v/v) TFA in water. Core histone purification used a gradient from 40% to 70% buffer B. Purified histones were aliquoted, lyophilized, and stored as powder at −70 °C.

**Histone octamer preparation.** Core histones were individually dissolved in an unfolding buffer (6 M guanidine HCl, 20 mM Tris pH 7.6, 10 mM NaCl, 1 mM EDTA, 1 mM DTT) and quantified by A280. Histones were combined at a 1:1:0.95:0.95 molar ratio of H2A:H2B:H3.2:H4. Pooled histones were diluted to a total concentration of approximately 1.0 mg/mL and dialyzed against refolding buffer (2 M NaCl, 10 mM Tris pH 7.6, 1 mM EDTA, 1 mM DTT) with three exchanges, each of which lasted for 6–12 h. The mixture was then recovered, concentrated to a volume of less than 1 mL on a 30 kDa centrifugal filtration concentrator, and cleared by centrifugation for 5 min at 17,000 × g at 4 °C. Supernatant was then resolved using a Superdex200 Increase 10/300GL on an AKTA FPLC column. Finally, octamers were analyzed for mass and purity by SEC and MALDI-TOF/TOF (Supplementary Fig. 7). Octamer-containing fractions were pooled, concentrated, and diluted 50% with glycerol before long-term storage at −20 °C.

**Nucleosome assembly.** Assembly reactions were performed at 2–5 μM (Widom601 147 bp and Gsc167 DNA, ThermoFisher) scale via serial salt dialysis. The histone octamer: DNA ratio used was optimized empirically to favor full conversion of free DNA into nucleosomes. All steps were performed at 4 °C. DNA, octamers, and buffer were combined to a final volume of 20 μL in buffer composed of 2 M NaCl, 10 mM Tris pH 7.6, 1 mM EDTA, 1 mM DTT and placed into 7000 MWCO Slide-A-Lyzer Mini dialysis buttons pre-moistened in 200 mL of initial buffer (1.4 M NaCl, 10 mM Tris pH 7.6, 0.1 mM EDTA, 1 mM DTT). After 1 h of dialysis in initial buffer, a peristaltic pump was used to add a total 350 mL of dilution buffer (10 mM NaCl, 10 mM Tris pH 7.6, 0.1 mM EDTA, 1 mM DTT) at a rate of 1 mL per min. Samples were moved to another 350 mL of dilution buffer 1–2 h after the peristaltic pump transfer was completed, and was allowed to dialyze in this new buffer for another 6–12 h. Samples were transferred to 300 mL of fresh dilution buffer and allowed to dialyze for a final 1–2 h before harvesting from the dialysis cassettes. After samples were removed from dialysis, they were then subject to centrifugation at 17,000 × g for 5 min to remove any precipitate that formed over the course of dialysis. Finally, the quality of nucleosome assembly was analyzed by native PAGE using 5% acrylamide, 0.5 X TBE gels. Nucleosomes of suitable quality were pooled and quantified by A260. Prior to affinity quantification we run qualitative EMSAs using a 40 μL mixture containing either 1 nM 5′ Cy5-labeled or non-modified DNA for Ethidium Bromide detection and a range of protein concentrations (indicated in the figures) in 1X binding buffer (10 mM Tris HCl pH7.5, 1 mM MgCl₂, 10 μM ZnCl₂, 10 mM KCl, 1 mM DTT, 5% (v/v) glycerol, 0.5 mg/mL BSA (Merck & Co., Inc.)) as described[63].

**Biolayer Interferometry.** was used to characterize binding kinetics between FoxH1 FH domain and either free or nucleosomal Gsc167 DNA on an Octet Red96e system (Sartorius). All reagents (DNA, nucleosomes, FoxH1) were exchanged into the following assay buffer for BLI experiments: 20 mM Tris pH 7.6, 100 mM NaCl, 2 mM DTT, 0.02% (v/v) Tween-20, 0.01% (w/v) BSA. Gsc167 or variant 601 nucleosomes were diluted to concentrations of approximately 5 ng/μL DNA. A two-fold

dilution series of FoxH1 was prepared, starting from a concentration of 125 nM and going down to 3.9 nM. Prior to experiments, streptavidin biosensors were pre-moistened and blocked in the assay buffer for at least 20 min. Binding kinetics experiments were performed in standard kinetics mode at 25 °C with the plate being rotated at 800 rpm throughout the assay. Loading of nucleosomes was limited to a threshold of 0.25 nm, analyte association was measured for 180 s, and dissociation for 240 s. To ensure consistent loading density on biosensors between nucleosome- and linear DNA-binding assays, the same sensors were used for both assays. Upon completion of nucleosome-binding experiments, octamers were dissociated from biosensors by washing 3x each for 15 s in octamer dissociation buffer (2.5 M NaCl, 20 mM Tris pH 7.6, 2 mM DTT, 0.02% (v/v) Tween-20, 0.01% (w/v) BSA). Octamer-depleted sensors were subsequently used for binding assays with linear DNA. Data analysis was performed using the Octet Data Analysis software, and data were fit to a 1:1 binding model for the estimation of kinetic parameters. $K_D$ data reported are derived from global fitting of five different protein concentrations, referenced against sensors with no protein added, with the standard error of the fit calculated by the analysis software (Octet® Analysis Studio, Sartorius). Goodness of fit was analyzed by visually examining residual plots for the fitted curves, as well as the $R^2$ and $X^2$ values of each fit.

### Electrophoretic mobility shift assay (EMSA)

Short duplex Cy®5-DNA was annealed using complementary single-strand HPLC purified DNAs. DNAs were mixed at equimolar concentrations (3 mM) in 20 mM Tris pH 7.0 at 25 °C and 10 mM NaCl, heated at 90 °C for 3 min and cooled down to room temperature for 2 h. Protein-DNA binding reactions were carried out for 30 min at 4 °C in 10 μL of binding buffer (100 mM Tris, 10% glycerol). A fixed concentration of 5′-end Cy5-labeled (Biomers, Germany) duplex DNA (7.5 nM) was incubated with increasing amounts of the different FoxH1 domain constructs. Electrophoresis was performed in native 6% polyacrylamide gels (1.5 mm thick), prepared with 19:1 40% acrylamide solution (PanReac AppliChem). The gels were run for 30 min in TG buffer at 150 V at 4 °C and exposed to a Typhoon imager (GE Healthcare).

For the NCP EMSAs, experiments were performed using identical buffers as for BLI. Binding reactions were loaded onto 0.5 X TAE 5% polyacrylamide gels with 10% sucrose, and samples were run at 100 V for 90 min at room temperature. Complexes were then visualized with ethidium bromide staining.

For the competition reactions of FL-FoxH1 with its FH domain, a fixed amount of 5′-end Cy5-labeled GG duplex DNA (0.7 ng), salmon sperm dsDNA (0.6 μg) and the FH domain were mixed and supplemented with increasing amounts of FL-FoxH1 cell lysate (0–76 μg) in 10 μL of binding buffer (200 mM TRIS pH 7.2, 100 mM NaCl and 2 mM TCEP).

For competition with FoxA2, a fixed amount of 5′-end Cy5-labeled TT duplex DNA (0.7 ng), salmon sperm dsDNA (0.6 μg) and the FoxA2 FH domain (43 ng) were mixed and supplemented with increasing amounts of FL-FoxH1 cell lysate (0–43 ng) in 10 μL of the same binding buffer described above. Each binding reaction was incubated for 30 min on ice, and then 10 μL of Orange G Loading Dye 2X was added to the mixture. Binding reactions were loaded into 1X TG native 6% polyacrylamide gel (1.5 mm thick), prepared with 19:1 40% acrylamide solution (PanReac AppliChem). The gels were run at 100 V for 1 h at 4 °C. Fluorescent image of the gels was acquired using the Thyphoon™ 6800 (Molecular Dynamics) with the Cy5 channel.

### Differential scanning fluorimetry assay

Differential scanning fluorimetry assays were performed using a LightCycler 480 real-time PCR device (Roche, Switzerland) as described previously. Protein (at a final concentration of 2.0 mg/ml) and SYPRO Orange (Sigma USA, at a final concentration of 5.0 mg/ml) were mixed and placed into the instrument at a heating rate of 1 °C/min. The fluorescence intensity vs. temperature (melting curve) was measured, and a melting temperature (Tm) was calculated from the maximum value of the first derivative of the curve using the program HTSDSF explorer (https://github.com/maciaslab/htsdsf_explorer)[64].

### Crystallization

The protein-DNA complexes were prepared by mixing protein with DNA at a 1:1.2 molar ratio. The FoxH1 complexes were prepared in buffer A (20 mM Tris-HCl pH 7.2, 100 mM NaCl, 10 mM potassium acetate, 2 mM TCEP) and the FoxA2 complexes in buffer B (40 mM Tris-HCl pH 7.2, 120 mM NaCl, 40 mM ammonium acetate, 20 mM magnesium chloride, 2 mM TCEP). The complexes were screened for crystal growth at the IBMB-IRB Barcelona Automated Crystallography Platform (PAC) using sitting-drop vapor diffusion in the SWISSCI 96-well format 3-lens plates with 25 μl of the reservoir solutions. All crystals grew within a few days. The conditions supporting the crystal growth were as followed:

*hFoxH1-GG complex* at 4.0 mg/mL protein concentration was mixed with 20% PEG 3350, 0.2 M ammonium chloride pH 6.3 reservoir solution at 200:100 nL reservoir to sample ratio; crystals grew at 4 °C; crystals were flash cooled and stored in liquid nitrogen using cryo-solution composed of 21% PEG 3350, 0.13 M ammonium chloride pH 6.3, and 18% ethylene glycol.

*dFoxH1-GG complex* at 4.0 mg/mL protein concentration was mixed with 35% PEG Smear Low* reservoir solution at 100:200 nL reservoir to sample ratio; crystals grew at 4 °C; crystals were directly flash cooled and stored in liquid nitrogen.

*dFoxH1-GT complex* at 4.0 mg/mL protein concentration was mixed with 41% PEG Smear Low* reservoir solution at 150:300 nL reservoir to sample ratio; crystals grew at 4 °C; crystals were directly flash cooled and stored in liquid nitrogen.

*dFoxH1-TT complex* at 4.5 mg/mL protein concentration was mixed with 20% PEG Smear High*, 0.1 M sodium acetate pH 4.5 reservoir solution at 100:200 nL reservoir to sample ratio; crystals grew at 4 °C; crystals were flash cooled and stored in liquid nitrogen using cryo-solution composed of 14% PEG Smear High*, 0.07 M sodium acetate pH 4.5, 18% glycerol, and 12% PEG 400.

*dFoxH1-TTAC complex* at 4.5 mg/mL protein concentration was mixed with 25% PEG Smear High*, 0.2 M lithium sulfate reservoir solution at 150:150 nL reservoir to sample ratio; crystals grew at 4 °C; crystals were flash cooled and stored in liquid nitrogen using cryo-solution composed of 17.5% PEG Smear High*, 0.14 M lithium sulfate, 18% glycerol, and 12% PEG 400.

*xFoxH1-GG complex* at 4.0 mg/mL protein concentration was mixed with 30% PEG 8000, 0.2 M ammonium sulfate, 0.1 M sodium cacodylate pH 6.5 reservoir solution at 100:200 nL reservoir to sample ratio; crystals grew at 20 °C; crystals were flash cooled and stored in liquid nitrogen using cryo-solution composed of 21% PEG 8000, 0.14 M ammonium sulfate, 18% glycerol, and 12% PEG 400.

*hFoxA2-TTAC complex* at 4.0 mg/mL protein concentration was mixed with 25% PEG 3350, 0.1 M Bis-Tris pH 6.5 reservoir solution at 150:150 nL reservoir to sample ratio; crystals grew at 4 °C; crystals were flash cooled and stored in liquid nitrogen using cryo-solution composed of 25% PEG 3350, 0.06 M Bis-Tris pH 5.5, and 12% glycerol.

*hFoxA2-TT complex* at 4.0 mg/mL protein concentration was mixed with 25% PEG 3350, 0.2 M ammonium sulfate, 0.1 M Bis-Tris pH 6.5 reservoir solution at 100:200 nL reservoir to sample ratio; crystals grew at 4 °C; crystals were flash cooled and stored in liquid nitrogen using cryo-solution composed of 25% PEG 3350, 0.12 M ammonium sulfate, 0.06 M Bis-Tris pH 5.5, and 12% glycerol.

*The PEG Smears are made by mixing PEG stocks (50% concentration) at an equal volume: PEG Smear Low is a mix of PEGs: 400,

500 MME, 600, and 1000; PEG Smear High is a mix of PEGs: 6000, 8000, and 10,000 without adding buffers.

## Data collection and structure determination

Diffraction data used for the structure determination were recorded at the ALBA beamline BL13-XALOC (Barcelona, Spain) and at the ESRF beamline ID30a3 (Grenoble, France). The data were processed, scaled, and merged with autoPROC[65] applying the anisotropy correction by STARANISO[66]. The CC1/2 criterion was used for selecting the diffraction resolution cut-off[67]. Initial phases were obtained by molecular replacement using PHASER[68,69] as part of the CCP4 and PHENIX suites (search model FoxK2, PDB code: 2c6y). REFMAC[70], phenix.refine[71], and BUSTER[72] were employed for the refinement, and COOT[73] for the manual improvement of the models. The PDB-REDO server was used for the selection of data resolution cut-off (paired-refinement), structure model optimization, and refinement with REFMAC[74]. All structures, as recommended by PHENIX and BUSTER documentation, and set as default by REFMAC5, have been refined with the riding hydrogens. However, only the two very high-resolution structures included the hydrogen atoms in the deposited files, while the others included a remark "REMARK 3 Hydrogens have been added in the riding positions". Optimization of target weights was done by the PDB-REDO server and automated optimization option in phenix.refine GUI. For the 0.98 Å or 1.19 Å structures, the best results were obtained with phenix.refine command line-based refinement dedicated to high resolution structures with the best weights being wxc_scale = 1.5 and 1.0 and wxu_scale = 1.5 and 1.0 for the 0.98 Å and 1.19 Å structures respectively. UCSF Chimera[75] and PyMol (Schrödinger) were used to prepare figures and calculate RMSD values for structural comparisons and Curves+[43] for DNA analysis.

## Reagents

A detailed list of reagents is provided in Supplementary Table 4.

## Reporting summary

Further information on research design is available in the Nature Portfolio Reporting Summary linked to this article.

# Data availability

The data that support this study are available from the corresponding author upon request. The atomic coordinates and structural factors have been deposited in the PDB with the following accession codes: hFoxH1-GG: 7YZB, dFoxH1-GG: 7YZ7, dFoxH1-GT: 7YZA, dFoxH1-TT: 7YZC, FoxA2-TTAC: 7YZE, dFoxH1-TTAC: 7YZD, FoxA2-TT: 7YZF, xFoxH1-GG: 7YZG. Source data are provided with this paper.

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

## Acknowledgements

The authors thank the Mass Spectrometry Core Facility (Universitat de Barcelona) and the PCB for support, the Protein Expression Core Facility

(IRB Barcelona) for providing reagents, and the Automated Crystallography Platform (IRB Barcelona-CSIC) and the ALBA Synchrotron for help with the experiments performed at the BL13-XALOC beamline. We also thank I. Benza, J. Cordero and J. Gordon for preliminary experiments, and B. Cánovas (IRB Barcelona) for helping us to prepare the experiments with HEK 293 T cell lysates. Our thanks also go to C.D. Lima (MSKCC) and the Tri-Institutional Therapeutics Discovery Institute for providing access to biolayer interferometry instrumentation. R.P. and B.B. are co-funded by the European Union's Horizon 2020 research and innovation programme under the Marie Skłodowska-Curie COFUND actions of IRB Barcelona and the PROBIST and PREBIST Postdoc and Predoc Programmes, respectively (agreements IRBPostPro2.0_600404, PROBIST_754510, and PREBIST_754558). N.A.P. received a Graduate Research Fellowship from the National Science Foundation 2017239554, NIH grant T32 GM115327-Tan and NIH grant F99CA264420. This work was supported by the Spanish MINECO program (BFU2014-53787-P, BFU2017-82675-P, and PID2021-122909NB-I00, M.J.M). Access to the ALBA Synchrotron was through BAG proposals 2018092972 and 2020094472. The work was also financed by the Spanish Ministry of Science and Education and the National Investigation Agency (MCIN/AEI/ 10.13039/501100011033), the European Regional Development Fund (ERDF) (BFU2014-53787-P and BFU2017-82675-P, M.J.M.), AGAUR (SGR-50, M.J.M.), and the NIH (grants R35-CA252978 (J.M.) and P30-CA008748 (MSKCC)). Y.D. is supported by the Josie Robertson Foundation, the Pershing Square Sohn Cancer Research Alliance, the NIH (CCSG core grant P30 CA008748, MSK SPORE P50 CA192937, and R35 GM138386), the Parker Institute for Cancer Immunotherapy (PICI), and the Anna Fuller Trust. In addition, the David lab is supported by Mr. William H. Goodwin, Mrs. Alice Goodwin, the Commonwealth Foundation for Cancer Research, and the Center for Experimental Therapeutics at MSKCC. We gratefully acknowledge institutional funding from the CERCA Programme of the Government of Catalonia, IRB Barcelona, the BBVA Foundation, and the Spanish Ministry of Science and Education through the Centres of Excellence Severo Ochoa Award. M.J.M. is an ICREA Programme Investigator.

## Author contributions

M.J.M. and J.M. designed the project. R.P. collected X-ray data, determined the structures and analyzed them with P.M.M. and M.J.M. E.A. cloned, expressed and purified all proteins and characterized their folding properties in solution. R.M. and L.R. performed the experiments with HEK 293 T cells. R.P. and B.B. screened crystallization conditions. E.A., L.R., and B.B. optimized the in vitro reconstitution of FoxH1-601 NCPs for EMSAs, N.P. and J.R.F. optimized the in vitro reconstitution of native *Gsc* and Widom601 NCPs and analyzed the quantitative binding assays with Y.D. L.R. performed the DSF assays. All authors contributed ideas to the project. M.J.M. R.P. and P.M.M. wrote the manuscript with contributions from all the other authors.

## Competing interests

J.M. has shares in the company Scholar Rock. The remaining authors declare no competing interests.
