## [Peer Review File · Nature Communications]

Molecular basis for DNA recognition by the maternal pioneer transcription factor FoxH1Reviewers' Comments:

Reviewer #1:

Remarks to the Author:

In this manuscript, the authors determined a total of seven structures of DNA bound FoxH1 or FoxA2 from Human, Brachydanio and Xenopus. Two structures are of high (i.e., atomic or near atomic) resolutions of 0.98 Å or 1.19 Å. In theory these datasets should provide significant amount of information about protein-DNA interactions. However, I don't think the authors took advantage of the atomic-resolution data by including hydrogen atoms in the refinement, and considering whether the geometric parameters of the protein and DNA were constrained or restrained during the refinement. The authors will have to explain how they treated the X-ray data sets at 0.98 Å differently from those at 2.82 Å. There are many protein-DNA structures at 2.8 Å resolution, but few at 1.0 Å or higher at the PDB data bank. At 1.0 Å resolution, every atom can be visualized, and a comparison with the protein-DNA complex structures at equivalent resolution should be made. It will be a pity if the authors do not analyze their atomic resolution structures adequately.

It is unclear which resolution was described in the paper. Instead of using "high resolution maps", just stated what the resolution is.

It is hard to follow when the same base pair of DNA was labeled with different number. For example, A1 and T1 are not of the same base pair, and the readers have to know that G6 and C11 form a base pair.

Avoid using unnecessary and unconventional abbreviations such as dHBs. In the case of atomic resolution structure, the hydrogen atoms involved in the H-bond could be easily visualized as well as the states of protonation or deprotonation of the Asp residue involved in direct interaction with DNA base. What were the pH values for PEG Smear Low and Smear High reservoir solution? Does the pH correlate with the Asp protonation?

The statement about the use of acidic residues (Asp and Glu) in DNA interaction does not reflect the current knowledge on the topic. Asp and Glu have been characterized as distinguishing cytosine and methylated cytosines, for example, see PMID: 23352388 or PMID: 30031306 and references therein.

In summary, the manuscript has potential, but the atomic resolution structures need to be fully and properly analyzed.

Reviewer #2:

Remarks to the Author:

The manuscript is focused on structural characterization of the FoxH1 transcription factor and the details of how it interacts with the DNA. The authors use X-ray crystallography, biochemistry, and extensive mutagenesis analysis to characterize various complexes of FoxH1 itself, FoxA2 and DNA templates. The authors have done a very thorough work and characterized the interactions of FoxH1 with the DNA in the greatest of detail. The resolution of the reported structures is also very high, which will provide a good resource for the researches interested in Fox family transcription factor interaction with the DNA.

The part about the nucleosome-FoxH1 interaction was intriguing. However, this most interesting to the general reader part is not as strong as the first structural part. While I do not question the idea that most Fox factors act as pioneer transcription factors, the experimental evidence in this particular case is not convincing enough.

If the nucleosome-FoxH1 part of the story was stronger, I would be positive about the paper. It would be important to show exactly where and how FoxH1 interacts with the nucleosome especially taking into account the difference in DNA geometry on the nucleosome. In the current state however I find it

more interesting for a more specialized reader, and in need of improvements for the nucleosome part.

In addition, I have a few more detailed comments:

- In order to make the protein-nucleosome interaction analysis more comprehensive, together with the Biolayer Interferometry shown in Fig. 6d, it is important to also show corresponding EMSAs. EMSAs are visual and also show the quality of the nucleosomal samples and potential heterogeneity. Please, include EMSAs for the 167 bp free-DNA/nucleosomes + FoxH1.
- Absolutely necessary is a negative control for Fig.6A and B: EMSA with 147bp Widom601 without a motif + the FoxH1 factor. Currently it is unclear why H1 linker histone was used as a control. Very often Widom sequence itself contains partial binding sites for TFs, so this has to be carefully checked. One other important control would be – EMSAs with free DNA 147bp/167bp (Widom and Widom+motif) + FoxH1
- Another important point – the schematic of the nucleosome with an exact linear position of the DNA should be added to the Fig.6, and the position of the motif marked. I can easily expect that that motif will be located Outside of the nucleosome region, and thus will behave almost like free DNA. Especially since a non-Widom sequence (weak octamer positioning) is used – what is often observed is that the octamer is positioned not as one expected and the flanking regions are different from predicted ones. One has to prove that the nucleosome is positioned like you anticipate, with an enzymatic digestion experiment. So at the moment I am not convinced that the FoxH1 binding site is positioned anywhere near the nucleosome core, and could easily be completely in the flanking region, if the octamer is positioned “on the edge” of the designed DNA sequence.

Minor Comments:

- Please, make the first paragraph of introduction more general. It is somewhat disconnected from the rest
- Please make sure that you explain all abbreviations (especially proteins) in the introduction, otherwise it becomes quite confusing for the reader.
- Fig 1e: why are there two bands (~free DNA)? Please comment. Was there a problem with primers? Is the DNA concentration than a correct one? 7.5 mM or would that be essentially lower, if the annealing was maybe 50% effective only ?

Reviewer #3:

Remarks to the Author:

The authors addressed a very interesting and important question: the molecular function of pioneer factors, in this case: how treh highly transcription factor FoxH1 binds to its target DNA. To do so, they generated an impressive selection of high-resolution crystal structures for DNA-bound FoxH1-FH-domains and they performed a series of in vitro assays to define functional relevant of specific domains for high-affinity FH-DNA/Nucleosome interaction. The approaches they took appear convincing and straight forward. However, at the end, the results remain descriptive and leave open the important question of in vivo relevance. In this context it is also important to note that in vivo affinity and possibly also specificity of FoxH1-DNA/Chromatin interaction might strongly depend on the interaction of activated Smad2/3 proteins and that studies using only an extended FH-domain bear a high risk of missing relevant parameter (see e22474. DOI: 10.7554/eLife.22474). Therefore, at least at some point also full-length FoxH1 protein functions should be analyzed and compared with those of the extended FH-domains.

Specific comment

To define a FoxH1 protein region combining DNA-binding functionality with options for efficient protein purification the authors generated different constructs and tested the purified protein in DNA/Protein interactions assays. Even if only a few of these constructs ultimately proved to be useful, their analyses revealed high affinity binding of an ext \rightarrow -ended FH-core construct to all three tested DNA-consensus versions (GG>GT>TT). It was also noted that a construct lacking the Wing2 domain binds the GG-consensus motif with very low affinity. Functional relevance of the Δ Wing2 construct appears questionable since it lacks >20% of the conserved amino acids. In view of the later shown detailed structural information (see Fig.2e, 3), it would have been interesting, important and highly beneficial to test mutations in individual relevant amino acids to confirm functional relevance (similarly as it was done for the KYR-motif, see Fig. 6e).

Building on the primary protein characterization the authors decided to use the conserved extended FH-domains from human, frog and fish FoxH1 for detailed structural analyses. The remarkable collection of crystal structures for DNA-bound FH-domains provides exiting insights into the molecular interactions not only of the core-FH domain but also on the role of the conserved flanking motifs including Wing2.

As based on structural similarities to linker Histones the authors next tested for interaction of the FH-domain with two different reconstitutes mononucleosomes containing FoxH1 consensus motifs. A corresponding interaction of FoxH1 with Histon-bound DNA is considered to support the idea of FoxH1 acting as a pioneer factor in early embryonic genome activation. Using EMSA and BLI approaches they indeed find that FH-domains from FoxH1 but also from FoxA1 interact with higher affinity to reconstituted nucleosomes than to linear DNA. In addition they show that the conserved KYR positions of FohH1 contribute to the high affinity binding. While the data are convincing, their significance is difficult to judge without corresponding control experiments with nucleosomes contained mutated and displaced FoxH1 consensus motifs. Further, at this point it would be important to include studies on full-length Foxh1 to confirm relevance of their extended FH-models.

Finally, the authors use public assessable Xenopus ChIP-seq data for FoxH1 and FoxA2 for bioinformatic analyses and they hypothesize about potential interactions (or binding competition) between FoxH1 and FoxA2. While this part is potentially interesting, the data shown are cryptic and conclusion appear highly speculative. Their hypothesis seems to be mostly based on a single parameter: the relative increase of enrichment for specific FoxH1-motifs (GK) in FoxH1 peaks between stage 8/9 and stage10,5, thus before and after onset of FoxA2 expression. A more detailed bioinformatic analyses also providing detailed information on the dynamics and statistics of FoxH1 occupancy, co-localization of FoxH1 and FoxA2 motifs (what about the other FoxA Proteins), and functional evidence for interaction should be provided.

Minor comments:

- Line 155 vs Line 266: Inconsistent numbers for DNA-binding affinities for Δ wing2:: `..a ~100-fold reduction...', `..bind DNA with 200 times lower affinity..'
- Line265/6: `..constructs lacking the C-terminal extension..' Why using plural, it seems only one construct was tested according to the text and Fig. 1?
- Line 406: bracket missing
- Lines 409-420: the usage of TT instead of TK (see Sup. Fig. 6) is misleading
- Lines 554-56: Very limited information on the protein-expression construct is provided
- Line 760: bracket missing
- Line 946: what is a `..Histone H1a FH domain..' ?

Reviewer #4:

Remarks to the Author:

This manuscript by Pluta et al. describes the structure of the forkhead (FH) domain transcription factor Foxh1. The structures of Foxh1 bound to multiple different sequence motifs provides valuable insights into how it recognizes DNA elements differently than other FH factors, most notably Foxa2. While these analyses are solid and important contributions, there are other aspects of this study that seem to be less conclusive. These issues are discussed in detail in the criticisms below.

Major criticisms

1. While the motif positions in Suppl Fig1b relative to nucleosome centers suggests that Foxh1 might bind DNA internal to the NCP, suggestive of a PF like FoxA, all the experiments used binding sites that are positioned at the nucleosome "edge". In Gsc 167, the site appears to be half off the nucleosome (if the opposite end of the DNA is the beginning of a 147bp core). Therefore "breathing" of the DNA on and off the nucleosome might permit access of Foxh1. Can Foxh1 bind to sites located more internal to a 147bp NCP? Perhaps this is addressed on lines 345-352 where the Widom nucleosome is discussed, but if so, this needs clarification. The text says that a 147bp nucleosome was constructed with Foxh1 site "close to the edge", but the manuscript lacks the details needed to understand what exactly this means: Figure 6 and Suppl Table 2 are cited but I can't locate the relevant information there.
2. The data shown in Suppl Fig1b and Fig6c are derived from MNase-seq experiments performed on various adult mouse tissues (brain, kidney, muscle, liver and heart) that don't express Foxh1 (see human UCSC GTEx v8 track). Therefore, it isn't clear whether nucleosomes positioning in these organs (containing mixed cell types) would be relevant to Foxh1 function in early embryonic epiblast. It may instead be that Foxh1 only binds to nucleosome-free sites (e.g., within ATAC-seq peaks). To conclude that Foxh1 sites are nucleosome bound would require MNase-type data derived from early embryos and not later stage tissues.
3. The analysis of Foxh1 versus Foxa2 binding presented in Suppl Fig 6 is not only difficult to understand as written, but also not very conclusive in demonstrating that the observations from *Xenopus* show that Foxa2 substitutes for Foxh1 in a "hand off". Foxa2 binding preferentially to TT motifs whereas Foxh1 remains bound to GK motifs seems to fit their model nicely. But the bioinformatic analysis on Foxh1 and Foxa2-bound regions doesn't really answer how this works. An experiment that would strongly support their model would be to use either naked DNA or NCPs that contain identical DNA sequences except for single TT or GG/GT motifs. Pre-bind these to Foxh1 and then add increasing concentrations of Foxa2 and examine binding behaviors. Based on the authors' work, one would expect that TT motifs would show that Foxa2 can compete with Foxh1 while not on identical GK-containing DNA.

Minor criticisms

1. Line 59 cites 2 papers that (Landsberger; Bogdanovic) seem irrelevant to the sentence. These are also cited on line 73, where they do seem appropriate to the material.
2. Line 90 should cite where the Foxh1 motifs shown are coming from: Chiu et al., 2014; Charney et al., 2017; Aragon et al 2019.
3. Line 97: please cite Dai et al 2021 review for statement that many FH domain structures bound to TT sites have been solved.
4. *Brachydanio* is an outdated name. The field has been using genus *Danio* for many years.
5. Line 271 refers to Suppl Fig3a, b for information about Wing1 binding to different sites but that is not what is shown in these panels. Perhaps the authors meant main Fig3a, b?
6. MNase data in Fig6c: It's not clear from the text whether the peaks shown are the straight MNase-seq reads. What exactly are we looking at? How the MNase data was used isn't explained and in Suppl Fig1b it's not at all clear how the authors obtained "distance from NCP centers." In 6c, each tissue shows a slightly different pattern. So was all the data equally sampled, then concatenated and average NCP centers were determined? Or some other approach was taken? No explanation is given on the bioinformatic treatment of these datasets.
7. Sentence on lines 406-410 is confusing. A period after "binds" to break sentence into two changes meaning to what I suspect the authors intended.
8. Suppl Fig1c: It's not clear what each bar represents as there are no labels. Legend says "the

presence of the same four DNAs shown in Figure 1d", but Suppl Fig1c has many bars that are different colors than in the main figure. Also, Suppl Fig1d heading doesn't seem to match what is shown in the panel.

9. In the end it's not clear to me whether the non-FH elements of DNA contact play any role in sequence selection for binding. Some non-FH interactions occur with nucleobases and not just phosphates/sugars, but do these only function in binding affinity to DNA? Would any nucleobase suffice in these positions for such interactions or do these AAs confer anything to sequence-specificity of binding (I'm not referring to TT versus GK, but instead more broadly to Fox sites versus random DNA sequence)?

10. An entire section was devoted to the findings of Hoogstein base pairing for binding to one motif but not the others. This raises more questions than it does provide answers. Is this in vivo relevant or important? Perhaps this section can be shortened and incorporated into another section on the general description of the binding interactions? Not sure it deserves its own section considering it doesn't add much insight.

General comments to reviewers

Thank you for these constructive comments that we believe have contributed to improving our manuscript. Following these recommendations, we have expanded the analysis of the structures to better balance the atomic information provided by the structures with the functional relevance of the complexes. We have included new sections in the text and the new figures to help illustrate this analysis. We have also expanded the section related to the analysis of FoxH1 interactions with NCPs.

REVIEWER COMMENTS

Reviewer #1 (Remarks to the Author):

In this manuscript, the authors determined a total of seven structures of DNA bound FoxH1 or FoxA2 from Human, Brachydanio (Danio) and Xenopus. Two structures are of high (i.e., atomic or near atomic) resolutions of 0.98 Å or 1.19 Å. In theory these datasets should provide a significant amount of information about protein-DNA interactions.

Query 1: However, I don't think the authors took advantage of the atomic-resolution data by including hydrogen atoms in the refinement, and considering whether the geometric parameters of the protein and DNA were constrained or restrained during the refinement.

Reply: In the previous version of the manuscript, all the reported structures - regardless of the resolution- were refined with hydrogens. During interactive refinements, riding hydrogens were either added manually (PHENIX and BUSTER) or by default (REFMAC5) and removed afterwards during refinement by PDB-REDO/REFMAC5 (default setting). Regarding the inclusion of the riding hydrogens in the deposited structures, there seems to be a lack of agreement within the community of crystallographers, and among software developers, even for high-resolution structures.

Following the recommendation of the reviewer, we decided to keep the riding hydrogens in the two high resolution structures. We delved into the PHENIX refinement documentation for high-resolution structure refinement. We found out that the command line-based instructions gave better results than the automated PHENIX GUI- and PDB-REDO-based ones, resulting in an improvement of the R-factors. The PDB/DCC-validated R-factors improved from 0.148/0.159 to 0.135/0.143 for the 0.98 Å structure and from 0.146/0.163 to 0.146/0.154 for the 1.18 Å one. The structures that have been re-refined include riding hydrogens have been re-deposited to substitute the previous ones.

The methods section has been expanded in the revised manuscript to include more details of the refinement steps and how protein and DNA were treated during the refinement. The protocols used were similar to those followed in previous protein-DNA complexes determined in our lab, (updated to the software improvements): Brucet et al, 2007, DOI:10.1074/jbc.M700236200), Martin Malpartida et al, 2017 (DOI:10.1038/s41467-017-02054-6), Aragon et al 2019,

Query 2: The authors will have to explain how they treated the X-ray data sets at 0.98 Å differently from those at 2.82 Å. There are many protein-DNA structures at 2.8 Å resolution, but few at 1.0 Å or higher at the PDB data bank. At 1.0 Å resolution, every atom can be visualized, and **a comparison with the protein-DNA complex structures at equivalent resolution should be made.** It will be a pity if the authors do not analyze their atomic resolution structures adequately.

Reply: As explained above, in the revised manuscript, we have expanded the section describing the two structures at 0.98 and 1.18 Å resolution in more detail. We have also referred to other TF complexes bound to DNA determined below 1.0 Å resolution and mentioned that the structure at 0.98 Å is the second-best complex of a TF bound to DNA determined to date.

Query 3: It is unclear which resolution was described in the paper. Instead of using “high resolution maps”, just state what the resolution is.

Reply: The resolution has now been included in the manuscript and in the figure panels.

Query 4: It is hard to follow when the same base pair of DNA was labeled with a different number. For example, A1 and T1 are not of the same base pair, and the readers have to know that G6 and C11 form a base pair.

Reply: We appreciate the reviewer’s point and understand its advantages for the ease of reading. However, in this case, we prefer to maintain the base numbering as presented in the deposited files. To indicate the directionality of each strand, we numbered them from 5’ to 3’ end as shown in the figures. In the text, the bases in the strand called chain B in the PDB file are numbered from 1 to 16, and the complementary strand (chain C) includes bases 1’ to 16’. We believe that the figures with the base numbering (e.g. Fig. 2) will help readers follow the text description. However, after giving some thought to this issue, we will follow the reviewer’s advice in future work.

Query 5: Avoid using unnecessary and unconventional abbreviations such as dHBs.

Reply: These abbreviations have been removed.

Query 6: In the case of atomic resolution structure, the hydrogen atoms involved in the H-bond could be easily visualized as well as the states of protonation or deprotonation of the Asp residue involved in direct interaction with DNA base.

Reply: The information we gathered related to the visualization of the hydrogen atoms indicates that unless truly sub-atomic resolution is achieved, then the vast majority of H atoms are not visible to X-rays (for example, see PMID:

19136630 or PMID: 26175905). Nonetheless, we evaluated the Asp protonation state in the 0.98 Å structure using several approaches, including Polder maps, but were able to observe only that the Asp residue is found in double conformation, with a minor occupancy of 0.25. We also found that the DNA backbone in proximity to the Asp residue also shows a double conformation to adapt to each conformation observed for the Asp side chain. These new findings are included in the re-refined and re-deposited structure, in the revised text and figures.

Query 7: What were the pH values for PEG Smear Low and Smear High reservoir solution? Does the pH correlate with the Asp protonation?

Reply: The PEG Smears we used, either bought from Molecular Dimensions in a form of the BCS (Basic Chemical Space) Screen or prepared in-house, do not include any buffering agent (they are PEGs dissolved in ultrapure 'Milli-Q' water), therefore the pH of the crystals of the GG and GT structures is that of the protein buffer (20 mM Tris-HCl pH 7.2, 100 mM NaCl, 10 mM potassium acetate, 2 mM TCEP). For the GG and GT crystals, the concentration of Tris-HCl pH 7.2 in the crystallization drop initiated at 13 mM and was increasing over time (until the crystals were directly flash frozen from their crystallization drops).

Query 8: The statement about the use of acidic residues (Asp and Glu) in DNA interaction does not reflect the current knowledge on the topic. Asp and Glu have been characterized as distinguishing cytosine and methylated cytosines, for example, see PMID: 23352388 or PMID: 30031306 and references therein.

Reply: Yes, this is the case here too. Asp binds to a Cytosine.

In summary, the manuscript has potential, but the atomic resolution structures need to be fully and properly analyzed.

Reply: We hope that this new version has been successful in describing the atomic resolution of some essential complexes and the power that these structures have in illustrating the functional mechanisms of FoxH1 and how it interacts with DNA.

Reviewer #2 (Remarks to the Author):

The manuscript is focused on structural characterization of the FoxH1 transcription factor and the details of how it interacts with the DNA. The authors use X-ray crystallography, biochemistry, and extensive mutagenesis analysis to characterize various complexes of FoxH1 itself, FoxA2 and DNA templates. The authors have done a very thorough work and characterized the interactions of FoxH1 with the DNA in the greatest of detail. The resolution of the reported structures is also very high, which will provide a good resource for the researchers interested in Fox family transcription factor interaction with the DNA. The part about the nucleosome-FoxH1 interaction was intriguing. However, this most interesting to the general reader part is not as strong as the first structural part. While I do not question the idea that most Fox factors act as pioneer transcription factors, the experimental evidence in this particular case is not convincing enough.

If the nucleosome-FoxH1 part of the story was stronger, I would be positive about the paper. It would be important to show exactly where and how FoxH1 interacts with the nucleosome especially taking into account the difference in DNA geometry on the nucleosome. In the current state however I find it more interesting for a more specialized reader, and in need of improvements for the nucleosome part.

In addition, I have a few more detailed comments:

Query 1: In order to make the protein-nucleosome interaction analysis more comprehensive, together with the Biolayer Interferometry shown in Fig. 6d, it is important to also show **corresponding EMSAs**. EMSAs are visual and also show the quality of the nucleosomal samples and potential heterogeneity. Please, **include EMSAs for the 167 bp free-DNA/nucleosomes + FoxH1**.

Reply: The NCP section of the manuscript has been thoroughly revised to include new panels and additional controls. We have included this EMSA as part of the new Figure 6.

Query 2: Absolutely necessary is a negative control for Fig.6A and B: **EMSA with 147bp Widom601 without a motif + the FoxH1 factor**. Currently it is unclear why H1 linker histone was used as a control. Very often the Widom sequence itself contains partial binding sites for TFs, so this has to be carefully checked. One other important control would be – **EMSAs with free DNA 147bp/167bp (Widom and Widom+motif) + FoxH1**

Reply: We used Histone H1 as a non-specific nucleosome protein binder since it binds to the nucleosomal linker (absent in the 147bp 601-NCP). We have prepared a Widom 601 sequence with the FoxH1 pseudo motifs removed (601m2). As expected, the 601m2-NCP does not interact with FoxH1 at the concentration ranges used in all binding assays. These controls and also those with the free DNAs (147 and 167bps) are now included in the new Figure 6 of the revised manuscript.

Query 3: Another important point – the schematic of the nucleosome with an exact linear position of the DNA should be added to the Fig.6, and the position of the motif marked.

Reply: We have included all sequences corresponding to the NCPs in the revised version of figure 6. We apologize for the absence of the FoxH1-601 sequence in the previous version.

Query 4: I can easily expect that that motif will be located **Outside of the nucleosome region**, and thus will behave almost like free DNA. Especially since a non-Widom sequence (weak octamer positioning) is used – what is often observed is that the octamer is positioned not as one expected and the flanking regions are different from predicted ones. One has to prove that the nucleosome is positioned like you anticipate, with an enzymatic digestion experiment. So at the moment I am not convinced that the FoxH1 binding site is positioned anywhere near the nucleosome core, and could easily be completely in the flanking region, if the octamer is positioned “on the edge” of the designed DNA sequence.

Reply: We have performed enzymatic digestion of the 167 Gsc as suggested. We found three enzymes that specifically cut the linear 167 Gsc sequence (new supplementary Figure 6). When the same enzymes are used with the NCP, only Stul (which site is located at the 5' end) is able to cut it. This result indicates that this area is exposed but not the others. Assuming the most extreme case, where the NCP is formed using the remaining bps, the FoxH1 site is still located in the supercoiled area of the NCP. We have included a new figure (supplementary Figure 6) to illustrate this scenario. We would like to mention that the affinity of FoxH1 for the linear Gsc DNA is 2.5-fold weaker than binding to the same DNA when wrapped around core histones. It is difficult to appreciate that binding to linear DNA protruding from the NCP is more favorable than binding directly to linear DNA, thereby suggesting that the FoxH1 site is located at the NCP even if it is at its edge. Moreover, we have used several Widom601-147bp NCPs to test FoxH1 interaction with NCPs that differ at the position of the FoxH1 site. According to our K_D s, it seems that the FoxH1 site can be located even at positions close to the dyad in this 147-NCPs. In these cases, the protein is still able to interact with the NCP at an affinity higher than to the linear DNA.

Minor Comments:

- Please, make the first paragraph of introduction more general. It is somewhat disconnected from the rest.
- Please make sure that you explain all abbreviations (especially proteins) in the introduction, otherwise it becomes quite confusing for the reader.

Reply: We have modified the introduction and made a list of abbreviations in the revised version of the manuscript. Thank you.

- Fig 1e: why are there two bands (~free DNA)? Please comment. Was there a problem with primers? Is the DNA concentration than a correct one? 7.5 mM or would that be essentially lower, if the annealing was maybe 50% effective only ?

Reply: We have repeated the annealing of the DNAs and we have included new EMSAs to replace the previous ones. These EMSAs have a single band corresponding to the double stranded DNA. The concentration is 7.5 nM.

Reviewer #3 (Remarks to the Author):

The authors addressed a very interesting and important question: the molecular function of pioneer factors, in this case: how treh highly transcription factor FoxH1 binds to its target DNA. To do so, they generated an impressive selection of high-resolution crystal structures for DNA-bound FoxH1-FH-domains and they performed a series of in vitro assays to define functional relevant of specific domains for high-affinity FH-DNA/Nucleosome interaction. The approaches they took appear convincing and straight forward. However, at the end, the results remain descriptive and leave open the important question of **in vivo** relevance. In this context it is also important to note that in vivo affinity and possibly also specificity of FoxH1-DNA/Chromatin interaction might strongly depend on the interaction of activated Smad2/3 proteins and that studies using only an extended FH-domain bear a high risk of missing relevant parameter (see e22474. DOI: 10.7554/eLife.22474). Therefore, **at least at some point also full-length FoxH1 protein functions should be analyzed and compared with those of the extended FH-domains.**

Specific comment

Query 1: To define a FoxH1 protein region combining DNA-binding functionality with options for efficient protein purification the authors generated different constructs and tested the purified protein in DNA/Protein interactions assays. Even if only a few of these constructs ultimately proved to be useful, their analyses revealed high affinity binding of an extended FH-core construct to all three tested DNA-consensus versions (GG>GT>TT). It was also noted that a construct lacking the Wing2 domain binds the GG-consensus motif with very low affinity. Functional relevance of the Δ Wing2 construct appears questionable since it lacks >20% of the conserved amino acids.

In view of the later shown detailed structural information (see Fig.2e, 3), it would have been interesting, important and highly beneficial to test mutations in individual relevant amino acids to confirm functional relevance (similarly as it was done for the KYR-motif, see Fig. 6e).

Reply: We have designed new point mutations to support the relevance of the two Wing regions in DNA binding. Mutations located at Wing1 (two Lys residues that participate in direct interactions with DNA) showed a decrease in DNA affinity, as expected based on the complexes. We have also introduced two point mutations and a double mutation in Wing2, at residues involved in long-range contacts. In contrast to the Δ Wing2 construct, which is still folded, all point mutations that we tried were prone to aggregation and precipitation during

purification. We have included a new EMSA (Figure 4e) showing the effects of point mutations in Wing1.

The mutations are highlighted in the FoxH1 sequence shown below:

GGKKKNYQRYPKPPYSYLAMIAMVIQNSPEKKLTLSEILKEISTLFPFFKGNKGYKWRDS
VRHNLSSYDCFVKVLKDPGKPPQGNFWTVEVNRIPLELLKRNQNTAVSRQDETIKFAQL
APYIFQG

Query 2: Building on the primary protein characterization the authors decided to use the conserved extended FH-domains from human, frog and fish FoxH1 for detailed structural analyses. The remarkable collection of crystal structures for DNA-bound FH-domains provides exciting insights into the molecular interactions not only of the core-FH domain but also on the role of the conserved flanking motifs including Wing2.

As based on structural similarities to linker Histones the authors next tested for interaction of the FH-domain with two different reconstituted mononucleosomes containing FoxH1 consensus motifs. A corresponding interaction of FoxH1 with Histone-bound DNA is considered to support the idea of FoxH1 acting as a pioneer factor in early embryonic genome activation. Using EMSA and BLI approaches they indeed find that FH-domains from FoxH1 but also from FoxA1 interact with higher affinity to reconstituted nucleosomes than to linear DNA. In addition, they show that the conserved KYR positions of FoxH1 contribute to the high affinity binding.

While the data are convincing, their significance is difficult to judge without corresponding control experiments with nucleosomes containing mutated and displaced FoxH1 consensus motifs. Further, at this point it would be important to include studies on full-length Foxh1 to confirm the relevance of their extended FH-models.

Reply: In the revised manuscript, we have included the control experiments using 601-NCP (147bp) with the FoxH1 site displaced to occupy three additional sites. These sites were selected based on the accessibility studies we performed using crystal structures of 601-NCPs and also taking into account a few TF-NCP complexes described in the literature. These experiments reveal that the FH domain interacts with all NCP variants at nanomolar affinity even with the FoxH1 site close to the dyad.

We have also compared the DNA binding capacity of the isolated extended FH domain and the FL protein expressed in HEK293T. The domain is able to recapitulate the DNA binding properties of the FL protein using a 16bp cy5-GG DNA.

Query 3: Finally, the authors use public assessable Xenopus ChIP-seq data for FoxH1 and FoxA2 for bioinformatic analyses and they hypothesize about potential interactions (or binding competition) between FoxH1 and FoxA2. While this part is potentially interesting, the data shown are cryptic and conclusion appear highly speculative.

Their hypothesis seems to be mostly based on a single parameter: the relative increase of enrichment for specific FoxH1-motifs (GK) in FoxH1 peaks between stage 8/9 and stage10,5, thus before and after onset of FoxA2 expression. A more detailed bioinformatic analyses also providing detailed information on the dynamics and statistics of FoxH1 occupancy, co-localization of FoxH1 and FoxA2 motifs (what about the other FoxA Proteins), and functional evidence for interaction should be provided.

Reply: This section has been modified to provide this information. We have also included competition assays between FoxH1 and FoxA2 proteins to highlight that FoxA2 (as an example of FOXA proteins) selects canonical forkhead motifs.

Minor comments:

- Line 155 vs Line 266: Inconsistent numbers for DNA-binding affinities for $\Delta wing2$: ‘..a ~100-fold reduction...’, ‘..bind DNA with 200 times lower affinity..’
- Line265/6: ‘..constructs lacking the C-terminal extension..’ Why using plural, it seems only one construct was tested according to the text and Fig. 1?
- Line 406: bracket missing
- Lines 409-420: the usage of TT instead of TK (see Sup. Fig. 6) is misleading
- Lines 554-56: Very limited information on the protein-expression construct is provided
- Line 760: bracket missing
- Line 946: what is a ‘..Histone H1a FH domain..’ ?

Thank you very much. We have included these corrections in the manuscript.

Reviewer #4 (Remarks to the Author):

This manuscript by Pluta et al. describes the structure of the forkhead (FH) domain transcription factor Foxh1. The structures of Foxh1 bound to multiple different sequence motifs provide valuable insights into how it recognizes DNA elements differently than other FH factors, most notably Foxa2. While these analyses are solid and important contributions, there are other aspects of this study that seem to be less conclusive. These issues are discussed in detail in the criticisms below.

Major criticisms

Query 1: While the motif positions in Suppl Fig1b relative to nucleosome centers suggests that Foxh1 might bind DNA internal to the NCP, suggestive of a PF like FoxA, all the experiments used binding sites that are positioned at the nucleosome “edge”. In Gsc 167, the site appears to be half off the nucleosome (if the opposite end of the DNA is the beginning of a 147bp core). Therefore “breathing” of the DNA on and off the nucleosome might permit access of Foxh1.

Query 2: Can Foxh1 bind to sites located more internal to a 147bp NCP? Perhaps this is addressed on lines 345-352 where the Widom nucleosome is discussed, but if so, this needs clarification. The text says that a 147bp nucleosome was constructed with Foxh1 site “close to the edge”, but the manuscript lacks the details needed to understand what exactly this means: Figure 6 and Suppl Table 2 are cited but I can’t locate the relevant information there.

Reply: In the revised manuscript, we have included the control experiments using 601-NCP (147bp) with the FoxH1 site displaced to occupy three additional sites, one of them with the motif centered as indicated in the new Figure 6. Based on the measured K_{DS} , which were all in the nanomolar range, it appears that FoxH1 recognizes its site even at positions close to the dyad.

We have also performed an enzymatic digestion assay of the 167 NCP Gsc. We found three enzymes that specifically cleave the linear DNA sequence (new supplementary Figure 6) but only Stu1 (which site is located at the 5' end) is able to cleave it when used in a nucleosome form. This result indicates that the Stu1 site is exposed whereas the other sites are protected. Assuming the most extreme case, where the nucleosome is formed using the remaining sequence (illustrated using a NCP structure in the supplementary Figure 6), the FoxH1 site is still located in the supercoiled area of the NCP. We would like to mention that the affinity of FoxH1 for the linear Gsc DNA is 2.5-fold weaker than binding to the same DNA when wrapped around core histones. It is difficult to appreciate that binding to linear DNA protruding from the NCP is more favorable than binding directly to linear DNA, thereby suggesting that the FoxH1 site is located at the NCP even if it is at its edge.

Query 3: The data shown in Suppl Fig1b and Fig6c are derived from MNase-seq experiments performed on various adult mouse tissues (brain, kidney, muscle, liver and heart) that don't express Foxh1 (see human UCSC GTEx v8

track). Therefore, it isn't clear whether nucleosomes positioning in these organs (containing mixed cell types) would be relevant to Foxh1 function in early embryonic epiblast. It may instead be that Foxh1 only binds to nucleosome-free sites (e.g., within ATAC-seq peaks). To conclude that Foxh1 sites are nucleosome bound would require MNase-type data derived from early embryos and not later stage tissues.

Reply: We selected these data sets precisely because we reasoned that in these tissues the nucleosome should be intact since FoxH1 and Gsc are not expressed, thus resembling the situation prior to FoxH1 binding to this promoter during embryonic development. We were intrigued by the agreement between the Gsc nucleosome boundaries observed in these data sets despite the difference in conditions studied. However, even if the boundaries are slightly different from those of true native conditions in which FoxH1 binds to the Gsc promoter, our experiments with the 147 NCP variants support the notion that the FoxH1 motif can be localized to different sites in a 147-NCP and the protein is still able to interact with them.

Query 4: The analysis of Foxh1 versus Foxa2 binding presented in Suppl Fig 6 is not only difficult to understand as written, but also not very conclusive in demonstrating that the observations from *Xenopus* show that Foxa2 substitutes for Foxh1 in a "hand off". Foxa2 binding preferentially to TT motifs whereas Foxh1 remains bound to GK motifs seems to fit their model nicely. But the bioinformatic analysis on Foxh1 and Foxa2-bound regions doesn't really answer how this works. An experiment that would strongly support their model would be to use either naked DNA or NCPs that contain identical DNA sequences except for single TT or GG/GT motifs. Pre-bind these to Foxh1 and then add increasing concentrations of Foxa2 and examine binding behaviors. Based on the authors' work, one would expect that TT motifs would show that Foxa2 can compete with Foxh1 while not on identical GK-containing DNA.

Reply: We have simplified the bioinformatics section accordingly. We have included a competition assay of FL-FoxH1 and FoxA2 using a cy5-labeled forkhead oligo in the revised version of the manuscript. We used FL-FoxH1 to benefit from the different size of this protein with respect to the FoxA2 FH domain. The results confirm the suggestion of this reviewer and that the two proteins efficiently bind to TT sites whereas FoxA2 binding to GG motifs is negligible and the domain is easily displaced by FL-FoxH1.

Minor criticisms

1. Line 59 cites 2 papers that (Landsberger; Bogdanovic) seem irrelevant to the sentence. These are also cited on line 73, where they do seem appropriate to the material.
2. Line 90 should cite where the Foxh1 motifs shown are coming from: Chiu et al., 2014; Charney et al., 2017; Aragon et al 2019.
3. Line 97: please cite Dai et al 2021 review for statement that many FH domain structures bound to TT sites have been solved.
4. *Brachydanio* is an outdated name. The field has been using genus *Danio* for many years.

5. Line 271 refers to Suppl Fig3a, b for information about Wing1 binding to different sites but that is not what is shown in these panels. Perhaps the authors meant main Fig3a, b?

Reply: Thank you for informing us about these mistakes. We have corrected them in the revised version of the manuscript.

6. MNase data in Fig6c: It's not clear from the text whether the peaks shown are the straight MNase-seq reads. What exactly are we looking at? How the MNase data was used isn't explained and in Suppl Fig1b it's not at all clear how the authors obtained "distance from NCP centers." In 6c, each tissue shows a slightly different pattern. So was all the data equally sampled, then concatenated and average NCP centers were determined? Or some other approach was taken? No explanation is given on the bioinformatic treatment of these datasets.

Reply: The data were taken from the databases without further analysis. We showed the "Nucleosome occupancy and position called by DANPOS2 " as defined by the authors.

As for the determination of the nucleosome center, we used the GSM2842982 MNaseSeq file. In this regard, we looked for local maxima using a python script that we wrote for this purpose (included below). These maxima were labeled as putative nucleosome centers. As the number of FoxH1 motifs analyzed was relatively small (21 promoters), we visually inspected each of them to validate the putative maxima as true nucleosome central positions. Distance from the center of the FoxH1 motif was then measured to the nearest nucleosome center. We have removed this analysis from the manuscript but we attach the script we wrote for the analysis.

```
import pyBigWig
import numpy

hard_thr=30
f = "GSM2842982_mm9_mouse_muscle_MNase-seq.bw"
bw = pyBigWig.open(f)
chroms=bw.chroms()
for key,value in chroms.items():
    for i in range(100,value-100,50):
        vals = numpy.array(bw.values(key, i, i+50))
        maxval=numpy.amax(vals)
        average=numpy.average(vals)
        maxpos = numpy.where(vals == numpy.amax(vals))
        for mymax in maxpos[0]:
            coord=mymax+i
            new_vals=numpy.array(bw.values(key, coord-60, coord+60))
            new_maxval=numpy.amax(new_vals)
            if bw.values(key,coord,coörd+1) [0]>=new_maxval and
new_maxval>hard_thr:
                print
(key+"\t"+str(coord)+"\t"+str(coord)+"\t"+str(bw.values(key, coord,coörd+1) [0]))

#now run bedtools sort and bedtools merge, and reduce band size to 1
```

7. Sentence on lines 406-410 is confusing. A period after "binds" to break sentence into two changes meaning to what I suspect the authors intended.

Reply: Thank you.

8. Suppl Fig1c: It's not clear what each bar represents as there are no labels. Legend says "the presence of the same four DNAs shown in Figure 1d", but Suppl Fig1c has many bars that are different colors than in the main figure. Also, Suppl Fig1d heading doesn't seem to match what is shown in the panel.

Reply: Thank you. We have corrected the mistake.

9. In the end it's not clear to me whether the non-FH elements of DNA contact play any role in sequence selection for binding. Some non-FH interactions occur with nucleobases and not just phosphates/sugars, but do these only function in binding affinity to DNA? Would any nucleobase suffice in these positions for such interactions or do these AAs confer anything to sequence-specificity of binding (I'm not referring to TT versus GK, but instead more broadly to Fox sites versus random DNA sequence)?

Reply: We have modified this section in the discussion. We have introduced point mutations at the protein level to characterize how some of these contacts are essential for overall affinity. It turns out that mutations introduced at residues that contact both minor grooves reduce overall protein affinity. However, since FoxH1 binds to various loci, it probably tolerates different bps at these positions, and some variants at positions 2-5 might even be balanced by the 3' site due to compensatory effects. Available data from sequence comparison and motif analysis indicate that there is a statistically significant propensity to select TGTGGATTG/C motifs. Given the very high affinities displayed by FoxH1 with respect to its preferred motifs, we could not detect affinity changes in EMSAs with DNAs that had slightly different modifications at each site of the motif.

10. An entire section was devoted to the findings of Hoogsteen base pairing for binding to one motif but not the others. This raises more questions than it does provide answers. Is this in vivo relevant or important? Perhaps this section can be shortened and incorporated into another section on the general description of the binding interactions? Not sure it deserves its own section considering it doesn't add much insight.

Reply: We have shortened the section describing the Hoogsteen bp accordingly.

Reviewers' Comments:

Reviewer #1:

Remarks to the Author:

The authors addressed my questions, and I have no further comments.

Reviewer #2:

Remarks to the Author:

The authors have addressed my comments. Now that the manuscript has improved I am positive about the publication

Reviewer #3:

Remarks to the Author:

All my concerns have been addressed - impressive work.

Reviewer #4:

Remarks to the Author:

The authors addressed most of my criticisms in a satisfactory manner and I recommend this paper for publication.